# The MBD7 complex promotes expression of methylated transgenes without significantly altering their methylation status

Dongming Li[1,2,3†], Ana Marie S Palanca[4†], So Youn Won[1†‡], Lei Gao[1,5], Ying Feng[1,6], Ajay A Vashisht[7], Li Liu[1], Yuanyuan Zhao[1], Xigang Liu[1,3], Xiuyun Wu[1,8], Shaofang Li[1], Brandon Le[1], Yun Ju Kim[1], Guodong Yang[1], Shengben Li[1], Jinyuan Liu[8], James A Wohlschlegel[7], Hongwei Guo[6], Beixin Mo[5], Xuemei Chen[1,9*], Julie A Law[4*]

[1]Department of Botany and Plant Sciences, Institute of Integrative Genome Biology, University of California, Riverside, United States; [2]School of Life Sciences, Lanzhou University, Lanzhou, China; [3]State Key Laboratory of Plant Cell and Chromosome Engineering, Hebei Collaboration Innovation Center for Cell Signaling, Center for Agricultural Resources Research, Institute of Genetics and Developmental Biology, Chinese Academy of Sciences, Shijiazhuang, China; [4]Plant Molecular and Cellular Biology Laboratory, Salk Institute for Biological Studies, La Jolla, United States; [5]College of Life Sciences and Oceanography, Guangdong Provincial Key Laboratory for Plant Epigenetics, Shenzhen University, Shenzhen, China; [6]State Key Laboratory of Protein and Plant Gene research, College of Life Sciences, Peking University, Beijing, China; [7]Department of Biological Chemistry, David Geffen School of Medicine at UCLA, Los Angeles, United States; [8]Laboratory of Molecular Biology and Protein Science, Laboratory of the Ministry of Education, Department of Biological Sciences and Biotechnology, Tsinghua University, Beijing, China; [9]Howard Hughes Medical Institute, University of California, Riverside, United States

*For correspondence: xuemei. chen@ucr.edu (XC); jlaw@salk.edu (JAL)

†These authors contributed equally to this work

Present address: ‡National Academy of Agricultural Science, Rural Development Administration, Suwon, Republic of Korea

Competing interests: The authors declare that no competing interests exist.

**Abstract** DNA methylation is associated with gene silencing in eukaryotic organisms. Although pathways controlling the establishment, maintenance and removal of DNA methylation are known, relatively little is understood about how DNA methylation influences gene expression. Here we identified a METHYL-CpG-BINDING DOMAIN 7 (MBD7) complex in *Arabidopsis thaliana* that suppresses the transcriptional silencing of two *LUCIFERASE* (*LUC*) reporters via a mechanism that is largely downstream of DNA methylation. Although mutations in components of the MBD7 complex resulted in modest increases in DNA methylation concomitant with decreased *LUC* expression, we found that these hyper-methylation and gene expression phenotypes can be genetically uncoupled. This finding, along with genome-wide profiling experiments showing minimal changes in DNA methylation upon disruption of the MBD7 complex, places the MBD7 complex amongst a small number of factors acting downstream of DNA methylation. This complex, however, is unique as it functions to suppress, rather than enforce, DNA methylation-mediated gene silencing.

## Introduction

DNA methylation is a highly conserved chromatin modification that is associated with gene silencing and plays critical roles in imprinting, transposon repression, and diverse developmental processes in eukaryotic organisms. In *Arabidopsis*, methylated regions of the genome can be separated into two main categories: (1) transposons and repeats that are heavily methylated at cytosines in all sequence contexts (namely CG, CHG and CHH where H indicates A, T or C), leading to gene silencing, and (2) body-methylated genes that harbor methylation exclusively in the CG context but remain highly expressed (*Lister et al., 2008*; *Cokus et al., 2008*). These global patterns of cytosine methylation reflect a balance between pathways controlling de novo methylation, maintenance methylation and demethylation (*Law et al., 2010a*; *Matzke and Mosher, 2014*).

De novo methylation of cytosines in all sequence contexts requires the RNA-directed DNA methylation (RdDM) pathway (*Law et al., 2010a*; *Matzke and Mosher, 2014*), which utilizes both 24-nucleotide small interfering RNAs (24-nt siRNAs) and long non-coding RNAs to target the de novo methyltransferase, DOMAINS REARRANGED METHYLTRANSFERASE 2 (DRM2), to repetitive regions of the genome. Production of both 24-nt siRNAs and non-coding RNAs requires specialized RNA polymerases (*Haag and Pikaard, 2011*; *Zhou et al., 2015*). Pol IV transcripts were recently identified (*Blevins et al., 2015*; *Zhai et al., 2015*; *Li et al., 2015a*) and these short non-coding RNAs are processed into 24-nt siRNAs in a one precursor, one siRNA fashion (*Blevins et al., 2015*; *Zhai et al., 2015*) and then loaded into the ARGONAUTE 4 (AGO4) clade of effector proteins. Pol V also generates non-coding RNAs at RdDM targets and these RNAs serve as a scaffold for the recruitment of many downstream RdDM factors, including siRNA-loaded AGO4 effector complexes and DRM2, ultimately leading to the deposition of DNA methylation and the establishment of gene silencing.

After the initial establishment of DNA methylation, several maintenance DNA methylation pathways are in place to ensure the faithful inheritance of DNA methylation patterns (*Law et al., 2010a*; *Matzke and Mosher, 2014*). Briefly, maintenance of CG methylation requires DNA METHYLTRANSFERASE 1 (MET1), whereas maintenance of CHG methylation (and some CHH methylation) relies on the activity of two related DNA methyltransferases, CHROMOMETHYLASE 2 and 3 (CMT2 and CMT3) (*Stroud et al., 2014*; *Zemach et al., 2013*). Finally, the remaining CHH methylation is maintained by DRM2 via the continuous action of the RdDM pathway.

Acting in opposition to the DNA methylation machinery are several DNA glycosylases, REPRESSOR OF SILENCING 1 (ROS1) and three paralogs: DEMETER (DME), DEMETER-LIKE 2 (DML2), and DEMETER-LIKE 3 (DML3). These glycosylases specifically remove methyl-cytosine bases, resulting in a net loss of DNA methylation (*Zhu, 2009*). While DME plays an essential role during plant reproduction, ROS1, DML2, and DML3 act in a semi-redundant manner to remove DNA methylation in vegetative tissue (*Lister et al., 2008*; *Zhu, 2009*; *Penterman et al., 2007*). These proteins tend to remove DNA methylation in regions that flank genes, and in *ros1 dml2 dml3* (*rdd*) triple mutants a subset of these genes are silenced, leading to a model in which ROS1, DML2 and DML3 function to remove DNA methylation and prevent transcriptional gene silencing (*Lister et al., 2008*; *Zhu, 2009*; *Penterman et al., 2007*).

Finally, to interpret the patterns of DNA methylation, there are two large families of methyl-DNA binding proteins in plants, both of which are conserved in mammals: the SET AND RING ASSOCIATED (SRA) domain family, and the METHYL-CpG-BINDING domain (MBD) family (*Defossez and Stancheva, 2011*; *Fournier et al., 2012*). While specific roles for several SRA domain proteins in the establishment and/or maintenance of DNA methylation in plants and mammals have been determined (*Johnson et al., 2007*, *2008*; *Woo et al., 2008*; *Bostick et al., 2007*; *Sharif et al., 2007*), roles for plant MBDs remain largely unknown. In mammals, MBD proteins function as part of large protein complexes associated with histone deacetylase and methyltransferase activities important for the establishment of repressive chromatin states (*Jones et al., 1998*; *Nan et al., 1998*; *Fuks et al., 2003*; *Zhang et al., 1999*). In *Arabidopsis*, there are 13 proteins that contain MBD domains (MBD1-13) (*Grafi et al., 2007*; *Zemach and Grafi, 2007*). Of these MBD proteins, early studies demonstrated that MBD5, MBD6 and MBD7 bind methylated DNA in vitro and localize to highly methylated, peri-centromeric regions of the genome in vivo, suggesting roles for these factors in gene silencing (*Zemach and Grafi, 2003*; *Scebba et al., 2003*; *Ito et al., 2003*; *Zemach et al., 2005*). A better understanding of how the MBD proteins bridge the gap between DNA methylation

and gene regulation, however, is just beginning to emerge. For example, MBD6 and MBD10 play roles in nucleolar dominance and rDNA silencing (*Preuss et al., 2008*), whereas MBD7 has been implicated in DNA demethylation based on genetic connections with the DNA demethylase, ROS1 (*Lang et al., 2015*; *Li et al., 2015b*; *Wang et al., 2015*).

Compared to the wealth of mechanistic detail regarding the proteins and pathways required to establish, maintain and even remove DNA methylation, relatively little is known about the events occurring downstream of DNA methylation. Previous genetic screens have identified *MORPHEUS MOLECULE 1* (*MOM1*) (*Amedeo et al., 2000*; *Won et al., 2012*), and *MICRORCHIDIA 1* and *6* (*MORC1* and *MORC6*) (*Moissiard et al., 2012*; *Lorković et al., 2012*; *Brabbs et al., 2013*), as genes that are required for the silencing of methylated loci but that do not control the levels of DNA methylation (*Amedeo et al., 2000*; *Won et al., 2012*; *Moissiard et al., 2012*; *Lorković et al., 2012*). In addition, mutations in *ARABIDOPSIS TRITHORAX-RELATED PROTEIN 5* and *6* (*atxr5* and *atxr6*) (*Jacob et al., 2009*) were also found to release gene silencing without significantly altering the pattern of DNA methylation. These findings suggest that there are factors that function downstream (or independently) of DNA methylation to facilitate gene silencing though unknown mechanisms. On the other hand, only one factor, SU(VAR)3–9 HOMOLOG 1 (SUVH1) (*Li et al., 2016*), has been identified to act downstream of DNA methylation to enable the transcription of methylated loci.

In the present study, we demonstrate a role for MBD7 and its associated proteins (LOW IN LUCIFERASE EXPRESSION (LIL) and REPRESSOR OF SILENCING 5 (ROS5), two Alpha Crystallin Domain (ACD) proteins, as well as REPRESSOR OF SILENCING 4 (ROS4)), in the suppression of gene silencing at methylated luciferase (*LUC*) reporter transgenes. In addition, we show that MBD5 co-purifies with a distinct set of ACD proteins but, unlike MBD7, is not required for *LUC* expression, suggesting different roles for the MBD5 and MBD7 complexes. To gain mechanistic insights into the role of the MBD7 complex in regulating gene expression, the DNA methylation and *LUC* expression levels at the reporter transgenes were determined in *mbd7*, *lil*, *ros4* and *ros5* mutants. These analyses revealed that members of the MBD7 complex function to promote luciferase expression without significantly altering DNA methylation levels. Furthermore, genome-wide characterization of DNA methylation patterns in *mbd7, lil*, and *rdd* mutants revealed little to no overlapping changes in DNA methylation. Together, these findings support a role for the MBD7 complex in the suppression of gene silencing that is primarily downstream of DNA methylation. While several proteins have been identified that reinforce gene silencing downstream of DNA methylation, the MBD7 complex is unique in its ability to overcome the silencing effects of DNA methylation, enabling the expression of several transgenes despite high levels of promoter methylation.

## Results

### Characterization of two luciferase (*LUC*)-based transcriptional gene silencing reporters

To identify genes that suppress transcriptional gene silencing in *Arabidopsis*, forward genetic screens were conducted using two *LUC*-based reporters, *LUCH* (*Won et al., 2012*) and *YJ* (*Li et al., 2016*), that were introduced into the *rdr6-11* mutant background to prevent post-transcriptional silencing. Here we compared the expression patterns and epigenetic features of the two *LUC* reporters, which differ in their basal levels of *LUC* expression despite harboring nearly identical transgenes that contain *LUC* genes driven by dual cauliflower mosaic virus *35S* promoters (*d35S*) (*Figure 1— source data 1*, *Figure 1—figure supplements 1A,B*). Given the known role of DNA methylation in regulating *LUC* expression at the *LUCH* reporter (*Won et al., 2012*), the DNA methylation and siRNA profiles for both *LUCH* and *YJ* reporters were determined by MethylC-sequencing (MethylC-seq) and small RNA sequencing (smRNA-seq), respectively (*Supplementary file 1A-D*), allowing either multi-mapping (*Figure 1—figure supplement 1C*) or unique reads (*Figure 1—figure supplement 1E*). At the *d35S* promoters, the two transgenes had similar patterns of 24-nt siRNAs and similar levels of DNA methylation in all sequence contexts (*Figure 1—figure supplement 1C,D*). However, in *LUCH*, DNA methylation extended beyond the *d35S* promoters into the *LUC* and *NPTII* coding regions (*Figure 1—figure supplement 1C,D*). One possible explanation for the difference in DNA methylation and *LUC* expression between *YJ* and *LUCH* reporters lies in the nature of the transgene insertions. Although both reporters segregate as single-copy insertions, analysis of the

MethylC-seq data supports the conclusion that the *YJ* and *LUCH* reporters represent single and multi-copy insertions into a single genomic locus, respectively (*Figure 1—figure supplement 2*). In addition to their copy number, these transgenes also differ in their integration sites, which may further contribute to their DNA methylation and expression profiles. Despite these differences, both reporters share a common feature that distinguishes them from most endogenous methylated loci: they contain genes that harbor high levels of promoter methylation but are not fully silenced. This feature makes these reporters well suited to screen for proteins that suppress gene silencing at methylated loci.

### *LIL* suppresses silencing at two *LUC* reporters

To identify factors that suppress gene silencing at the *LUC* reporters, two independent genetic screens using either the *YJ* or the *LUCH* reporter line, were performed. Two mutants with reduced

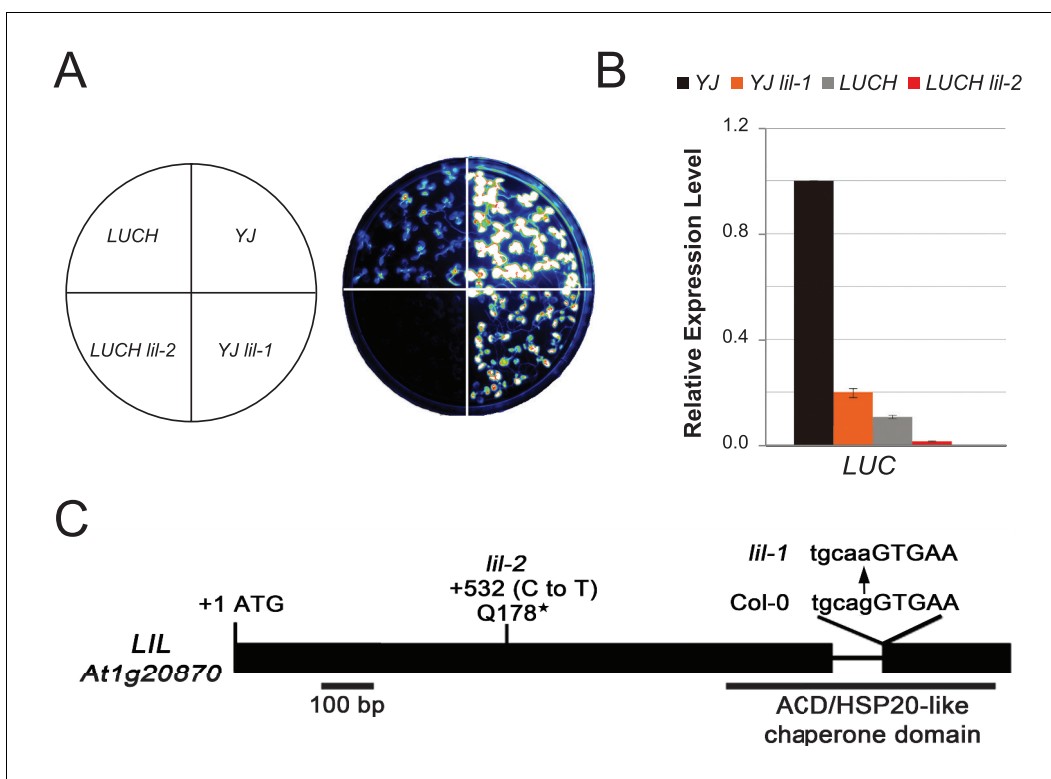

**Figure 1.** LIL promotes expression of methylated LUC reporters. (**A**) Luciferase (*LUC*) luminescence in *YJ*, *YJ lil-1*, *LUCH* and *LUCH lil-2* seedlings as diagramed on the left. (**B**) Quantification of *LUC* transcript levels by RT-qPCR. Transcript levels were normalized to *UBIQUITIN5* with the expression level of *LUC* in the *YJ* control set to one. Error bars indicate the standard deviation from two biological replicates. (**C**) LIL gene structure showing the positions of the two isolated mutations relative to the exons (black bars) and a single intron (black line). Lower and upper case letters represent intron and exon sequences, respectively. The region encoding the conserved ACD or HSP20-like chaperone domain is indicated below.

The following source data and figure supplements are available for figure 1:

**Source data 1.** Alignment of the *YJ* and *LUCH* transgenes.

**Figure supplement 1.** Characterization of the *LUCH* and *YJ* luciferase (*LUC*) reporter lines.

**Figure supplement 2.** Investigation of transgene copy number.

**Figure supplement 3.** Sequence alignment of LIL and its paralogs.

luciferase luminescence, one in each reporter background, were identified (*Figure 1A*) and RT-qPCR analysis confirmed reduced *LUC* expression in these mutant backgrounds relative to their respective controls (*Figure 1B*). Map-based cloning and candidate gene sequencing revealed that the same gene (*At1g20870*) was disrupted in both mutants. The alleles isolated from the *YJ* and *LUCH* screens are hereafter referred to as *lil-1* and *lil-2* (LIL, LOW IN LUCIFERASE EXPRESSION), respectively. The *lil-1* allele contains a G-to-A mutation, which disrupts the single splice acceptor site of *At1g20870* (*Figure 1C*). Although *lil-2* was isolated from a T-DNA mutagenesis screen, this allele does not contain a T-DNA insertion, but instead contains a C-to-T mutation that introduces a premature stop codon (*Figure 1C*). *LIL* is predicted to encode a 51.9 kDa protein containing a carboxy-terminal Alpha Crystallin Domain (ACD) or Heat Shock Protein 20 like (HSP20-like) chaperone domain (*Figure 1C*). Among the 25 *Arabidopsis* ACD-containing proteins, LIL and three paralogs form one clade (*Scharf et al., 2001*) (alignments shown in *Figure 1—figure supplement 3*). Two members of this clade, LIL (also known as INCREASED DNA METHYLATION 3 (IDM3) [*Lang et al., 2015*] or INCREASED DNA METHYLATION 2-LIKE 1 (IDL1) [*Li et al., 2015b*]) and ROS5 (*Zhao et al., 2014*)/ IDM2 (*Qian et al., 2014*) (At1g54840) were recently found to prevent DNA hyper-methylation and enable gene expression at other reporter transgenes and select genomic loci, demonstrating they also act to suppress gene silencing (*Lang et al., 2015*; *Li et al., 2015b*; *Zhao et al., 2014*; *Qian et al., 2014*).

## LIL is associated with MBD7

To gain insight into the function of LIL, a yeast two-hybrid (Y2H) screen was conducted to identify LIL-interacting proteins. Using full-length LIL as bait and an *Arabidopsis* cDNA library as prey, LIL was found to interact with three MBD proteins: MBD5, MBD6 and MBD7 (*Figure 2—figure supplement 1A,B*). To map the interaction domains of these proteins, select subdomains were subjected to additional Y2H assays (*Figure 2—figure supplement 1A,B*). Using either the N-terminal portion of LIL (LILN3) or the ACD/HSP20 domain alone (LILD1), no interactions were observed with the full length MBD proteins (*Figure 2—figure supplement 1A,B*). However, an interaction was detected between the last MBD domain of MBD7 (MBD7d3) and the full length LIL protein (*Figure 2—figure supplement 1A,B*). Previously, *Lang et al. (2015)* mapped the interaction domain between LIL and MBD7 to the C-terminal sticky-c (StkC) domain (*Zemach et al., 2009*) of MBD7 and found that the three MBD domains of MBD7 alone were not sufficient to mediate an interaction with LIL. However, the MBD7d3 construct used here represents an extended version of the third MBD domain that has minimal overlap with the StkC domain, suggesting that the interaction between MBD7 and LIL can be mediated by several regions of the MBD7 protein, namely, the StkC domain (*Lang et al., 2015*), as well as the last MBD domain (*Figure 2—figure supplement 1A,B*).

In a parallel effort, epitope-tagged versions of MBD5 and MBD7, expressed under the control of their endogenous promoters, were affinity purified and found to associate with distinct sets of ACD domain proteins by Mass Spectrometry (*Figure 2—figure supplement 1C,D*). Specifically, a 3x-Flag-tagged version of MBD5 (*pMBD5::MBD5-3x-Flag*) was shown to co-purify with ACD15.5 (At1g76440) and ACD21.4 (At1g54850) as well as low amounts of MBD6 (*Figure 2—figure supplement 1D*), while purification of a 3x-HA tagged version of MBD7 (*pMBD7::MBD7-3xHA*) revealed a specific interaction with LIL (*Figure 2—figure supplement 1C*). As MBD7 had previously been shown to also associate with ROS4, ROS5 (*Lang et al., 2015*; *Li et al., 2015b*; *Wang et al., 2015*), and HARBINGER TRANSPOSON-DERIVED PROTEIN 1 (HDP1) and HDP2 (*Duan et al., 2017*), additional purifications using a 3x-Flag-tagged version of MBD7 (*pMBD7::MBD7-3xFlag*) were conducted and these experiments yielded peptides corresponding to LIL as well as all the aforementioned MBD7-associated proteins (*Figure 2—figure supplement 1C*). Taken together, these findings demonstrate that MBD5 and MBD7 associate with different subsets of ACD proteins to form distinct MBD-ACD protein complexes, the roles and compositions of which are just beginning to be explored.

## *MBD7*–associated factors suppress silencing at the *LUC* reporter

To determine the roles of the MBD5 and MBD7 complexes in the regulation of gene expression, mutant alleles in the various components were obtained (*Figure 2—figure supplement 2*) and their effect on *LUC* expression was determined. For these experiments, two alleles of *ROS4* (*ros4-2* and

*ros4*-3) were recovered from the *YJ* reporter screen and the following additional mutant lines were crossed into the *YJ* reporter background in either the Col or L*er* ecotypes (see Materials and methods): *mbd5-1, mbd5-3, mbd7-3, mbd7-4, mbd7-5,* and *ros5-4*. The *mbd5* mutations had no effect on *LUC* expression at the *YJ* reporter (*Figure 2—figure supplement 3A,B*). Conversely, in the *mbd7, ros4,* and *ros5* mutants, *LUC* expression was reduced to similar levels as observed in *lil-1* (*Figure 2A,B*). In addition, expression of the *d35S*-driven *NPTII* gene present in both *YJ* and *LUCH* reporters was also reduced in these mutants (*Figure 2C*). These findings are consistent with the co-purification of these factors by Mass Spectrometry and demonstrate a role for the MBD7 complex, but not MBD5, in promoting the expression of the *LUC* reporter.

## The MBD7-LIL complex regulates *LUC* expression at the transcriptional level

Previous analyses have shown that both the *YJ* (*Li et al., 2016*) and *LUCH* (*Won et al., 2012*) reporters are regulated by DNA methylation such that treatment with the cytosine methylation inhibitor 5-aza-2′-deoxycytidine (5-Aza-dC) results in increased *LUC* expression. To determine if DNA methylation is necessary for the phenotypes observed upon disruption of the MBD7 complex, we assessed the effects of 5-Aza-dC treatment on *LUC* expression in several alleles of *lil* and *mbd7* and found the levels of *LUC* expression in these mutants were indistinguishable from their respective controls (*Figure 3A,B*). Thus, LIL and MBD7 are only necessary for the expression of the *LUC* reporters when they harbor DNA methylation, affirming their roles in the transcriptional, rather than post-transcriptional, regulation of *LUC* expression.

## Disruption of the MBD7 complex results in minimal effects on DNA methylation at the *LUC* reporters

To determine whether the decreased *LUC* expression observed upon disruption of the MBD7 complex is associated with an increase in DNA methylation, methyl-DNA cutting assays and MethylC-seq experiments were conducted. For the methylation cutting assays, genomic DNA was either mock treated or digested with the McrBC restriction enzyme followed by PCR amplification. Since McrBC specifically cuts methylated DNA, hyper-methylated targets are expected to exhibit reduced levels of PCR amplification. Using this assay, amplification of the *d35S* promoter was reduced in both *lil* mutants relative to their respective controls, indicating that mutations in *LIL* increased the level of DNA methylation at the *d35S* promoter (*Figure 4A*; gel based). Similar results were obtained for *lil-1, mbd7, ros4,* and *ros5* mutants when qPCR was performed to quantify DNA levels from mock or McrBC treated samples (*Figure 4B*; qPCR). Together, these analyses reveal a correlation between disruption of the MBD7 complex and an increase in DNA methylation at the *d35S* promoter driving *LUC* expression, but they do not reveal the location or extent of hyper-methylation.

To quantitatively determine the changes in methylation occurring across the entirety of the *LUC* reporters upon disruption of the MBD7 complex, the patterns of DNA methylation were determined at single nucleotide resolution by bisulfite sequencing using the MethylC-seq method (*Urich et al., 2015*) in *mbd7* and *lil* mutants (*Supplementary file 1A, 1B and 1E*). These transgene analyses were limited to the *lil-1* and *lil-2* alleles, which are point mutations, and the *mbd7-5* allele, which is the only *mbd7* allele where the T-DNA insertion does not share sequence homology with the *d35S* promoters present in the *LUC* reporters (*Figure 4C*, see 'no *YJ* transgene control tracks'). At the *LUC* reporters, quantification of the average methylation levels across the *d35S* promoter (282–1036 nt) revealed a surprisingly modest but consistent increase in DNA methylation in the CG context in *lil-2, mbd7-5,* and three biological replicates of *lil-1* (*Figure 4D*). Further inspection of the DNA methylation profiles revealed that this hyper-methylation is largely restricted to a small region at the 3′ end of the *d35S* promoter, which is within the region amplified in the McrBC-PCR methyl-cutting assays (*Figure 4C*; right panels). Quantification of DNA methylation at this region was assessed either in its entirety (923–1318), to best evaluate the change in methylation at the *LUCH* reporter, or in a more limited region (923–1004) that encompasses most of the methylation observed in the *YJ* reporter. In the *lil-2* mutant, a slight increase in CG and CHG methylation, but no change in CHH methylation, was observed at the *LUCH* reporter (*Figure 4D*; Full McrBC). For the *lil-1* replicates and *mbd7-5*, a more prominent increase in CG and CHG methylation was observed, and these DNA methylation defects were significantly reversed by introduction of an *MBD7-3xHA* construct that rescues the *LUC*

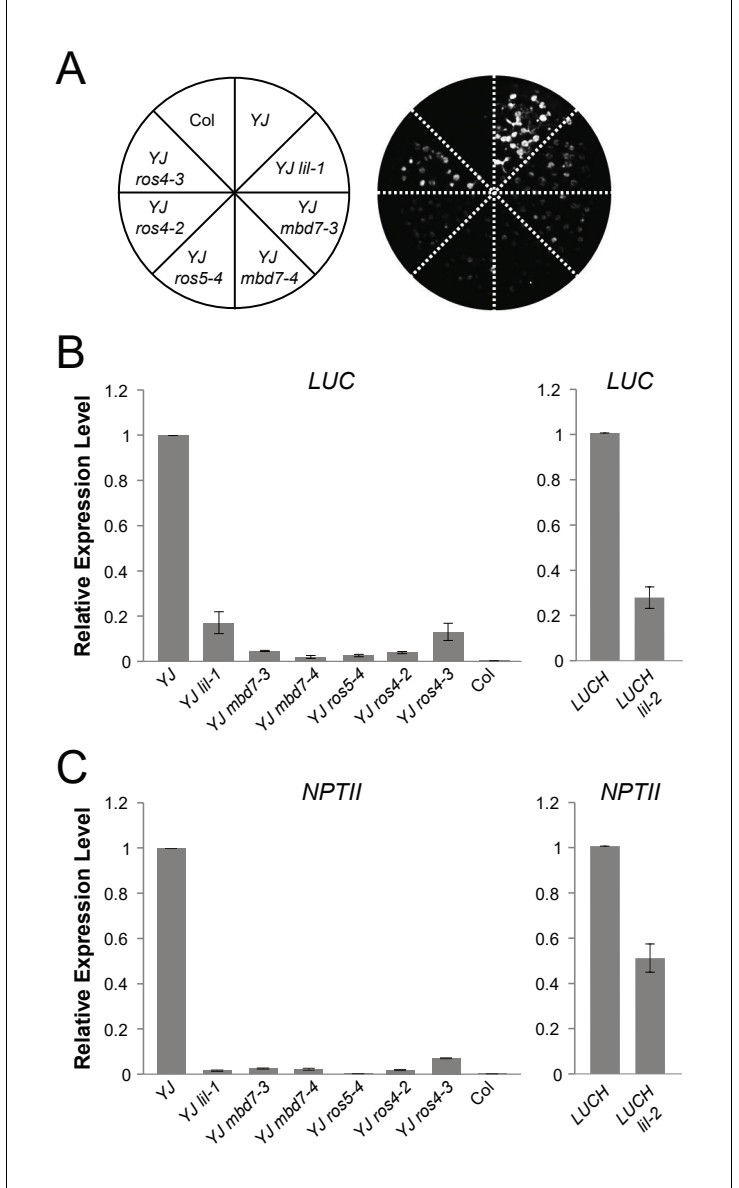

**Figure 2.** *mbd7*, *ros4*, and *ros5* mutants phenocopy *lil-1* at the YJ transgene. (**A**) Luciferase (LUC) luminescence in 10-day-old seedlings as diagramed on the left. (**B**) Quantification of *LUC* or (**C**) *NPTII* transcript levels by RT-qPCR. Transcript levels were normalized to *UBIQUITIN5* with the expression level of *LUC* or *NPTII* in the *YJ* or *LUCH* controls set to one. Error bars indicate the standard deviation from two biological replicates.

The following figure supplements are available for figure 2:

**Figure supplement 1.** Yeast two-hybrid and affinity purification analyses.

**Figure supplement 2.** Isolation and characterization of *ros4*, *mbd5*, *mbd7*, and *ros5* mutants.

**Figure supplement 3.** *mbd5* mutants do not exhibit reduced *LUC* expression.

expression phenotype at the *YJ* reporter (***Figure 4D*** short McrBC and ***Figure 4—figure supplement 1A***, respectively). As these findings reveal a correlation between the presence of a functional MBD7 complex and the DNA methylation status of the *YJ* reporter, several chromatin immunoprecipitation (ChIP) experiments were conducted to determine whether the MBD7 complex associates with this

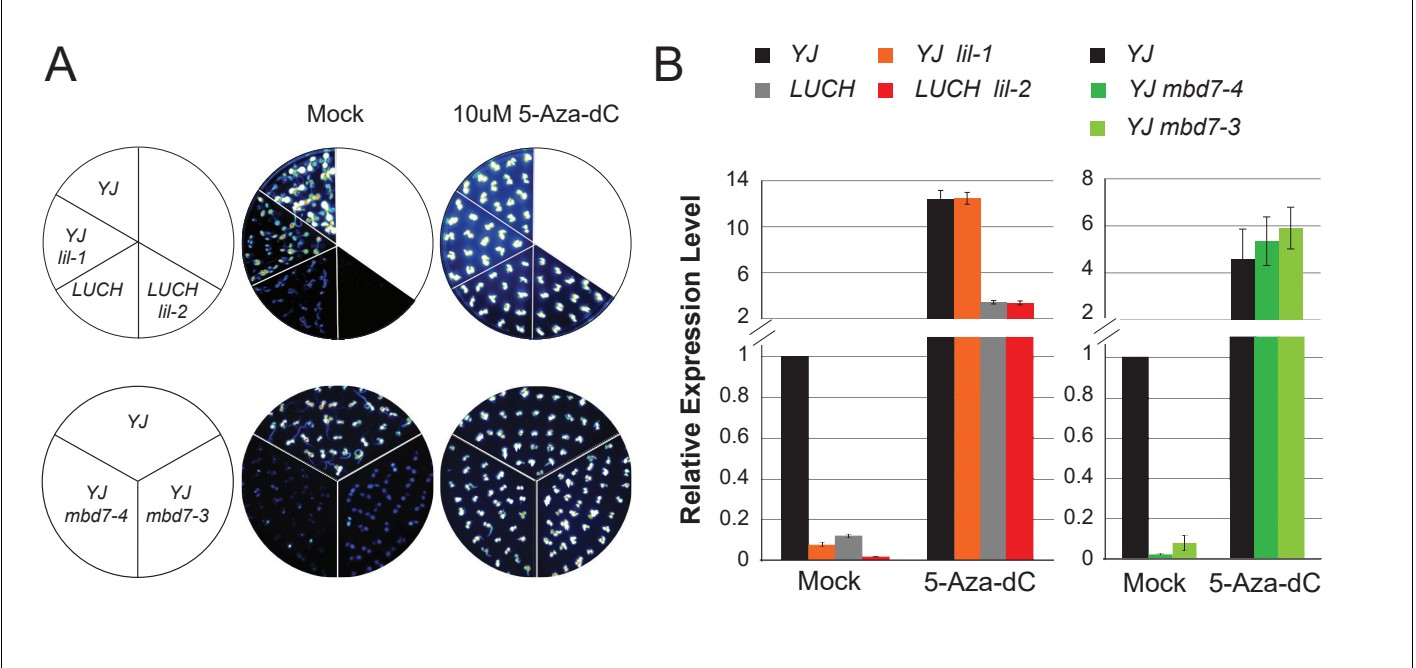

**Figure 3.** MBD7 and LIL regulate *LUC* expression in a DNA methylation-dependent manner. (**A**) Luciferase luminescence of mock or 5-aza-2'-deoxycytidine (5-Aza-dC)-treated seedlings as diagramed on the left. (**B**) Quantification of *LUC* transcript levels by RT-qPCR. *LUC* transcript levels were normalized to *UBIQUITIN5* with the expression level of *LUC* in the *YJ* control set to one. Error bars indicate the standard deviation from three biological replicates for the *mbd7* datasets and three technical replicates for the *lil* dataset.

reporter. While no enrichment was observed with either the 3xHA or 3xFlag tagged MBD7 proteins expressed under their native promoters, enrichment was observed at the *d35S* promoter using a previously characterized MBD7-GFP line driven by a *35S* promoter (*Wang et al., 2015*) (*Figure 4E*). Taken together, these methylation and ChIP analyses are consistent with the hyper-methylation phenotype observed in the methyl-cutting assays and suggest a direct role for the MBD7 complex in regulating expression at the *YJ* reporter.

## Genetic uncoupling of the hyper-methylation and *LUC* expression phenotypes at the *YJ* reporter

Given the limited changes in DNA methylation observed at the *LUC* reporters in the *lil* and *mbd7* mutants (especially compared to the striking reduction in *LUC* expression), we sought to better understand how alterations in DNA methylation influence *LUC* expression at the *YJ* reporter. We therefore manipulated the methylation pattern of the *d35S* promoter using known DNA methylation mutants and then assessed the effect on *LUC* expression. We chose two strong RNA-directed DNA methylation (RdDM) mutants, *ago4* and *nrpe1*, and the triple demethylase mutant, *rdd*. Unfortunately, the T-DNAs in these mutants have significant sequence homology with the *d35S* promoter in the *YJ* transgene (*Supplementary file 1A*). Thus, to specifically assess DNA methylation at the *d35S* promoter driving the *LUC* gene, without interference from the 94% identical *d35S* promoter driving *NPTII* expression (Figure 1—figure supplement 1C,E) or the similar *d35S* promoters present in the T-DNA insertion mutant backgrounds, traditional bisulfite conversion assays coupled with Sanger sequencing were conducted. For comparison, a full set of alleles in the Col ecotype (*lil-1*, *mbd7-4*, *ago4-6*, *nrpe1-11*, and *rdd*) and the complementation data set in the L*er* ecotype, all in the *YJ rdr6-11* background, were included.

First, we compared results from traditional bisulfite sequencing in the *lil* and *mbd7* mutants with those from MethylC-seq. The DNA methylation levels observed at the *35S* promoter by traditional bisulfite sequencing are represented in browser track format (*Figure 5A* and *Supplementary file 1D*: YJ_tradBS_WigTracks) and show similar patterns of methylation in the *lil* and *mbd7* mutants

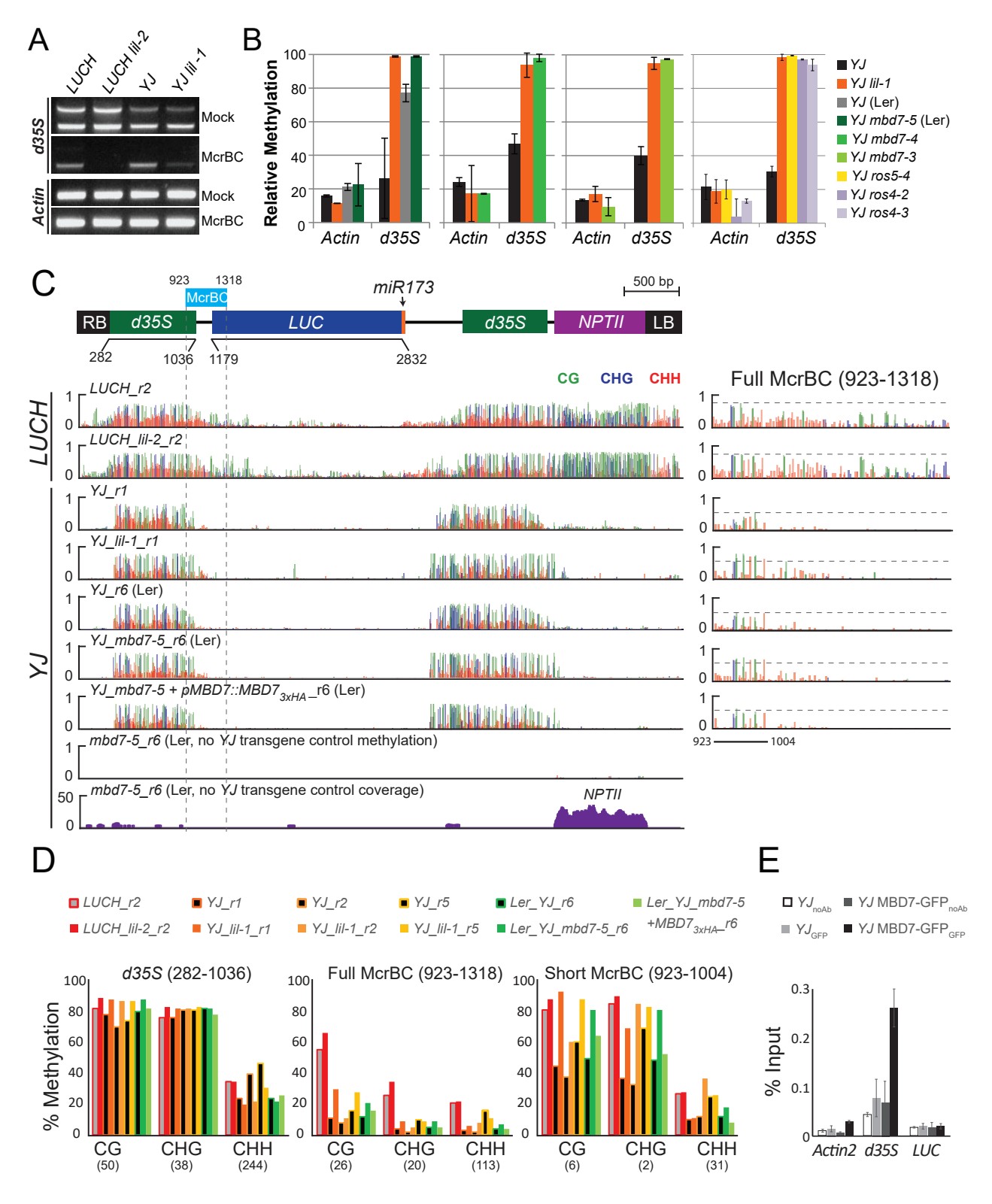

**Figure 4.** Disruption of the MBD7 complex results in subtle, but reproducible hyper-methylation at the *d35S* promoter. Cytosine methylation analysis by McrBC-PCR (**A**) and McrBC-qPCR (**B**) using mock or McrBC treated genomic DNA from the indicated genotypes. In (**A**), two tandem copies of the *35S* sequence result in two PCR bands. *ACTIN1*, which lacks methylation, was used as the internal loading control. In (**B**), the relative methylation levels are plotted and the error bars indicate the standard deviation from two biological replicates. (**C**) A diagram of the *YJ* reporter drawn to scale relative to

*Figure 4 continued on next page*

*Figure 4 continued*

the browser tracks shown directly below. The light blue box indicates the region amplified for the McrBC assay. This region is expanded in the far right panel. DNA methylation tracks show the DNA methylation level (0–1, where 1 = 100% methylated) in the CG (green), CHG (blue), and CHH (red) sequence contexts at cytosines covered by at least five reads. The *LUCH_r2* and *YJ_r1* tracks correspond to those shown in *Figure 1—figure supplement 1*. The *YJ mbd7-5 + pMBD7::MBD7-3xHA_r6* (Ler) line corresponds to the 'ins1a' line characterized in *Figure 4—figure supplement 1*. The coverage track for *mbd7-5_r6* (bottom) shows the number of reads (*y*-axis) mapped to the transgene, demonstrating that sequence homology between the T-DNA in the *mbd7-5* mutant and the LUC reporter is largely limited to the *NPTII* drug resistance gene. To facilitate visual assessment of the changes in DNA methylation, dashed horizontal lines are set relative to the maximal CG methylation at the 3' end of the *d35S* promoter in the control *LUCH* and *YJ* reporters. (D) Quantification of the average percent methylation across the entire *d35S* promoter (282–1036), the Full McrBC region (923–1318) or a Short MrcBC region (923–1004) within the *YJ* transgene in the lines presented in (C). Each sample is appended with an 'r#' to indicate samples that were processed and sequenced together. Identical genotypes with different r#'s indicate biological replicates (e.g., *YJ_r1*, *YJ_r2*, and *YJ_r5* are biological replicates). The number of cytosines in each sequence context within the quantified regions is indicated in parentheses. Note that the two *d35S* promoters driving *LUC* and *NPTII* are 94% identical in sequences, thus the DNA methylation data includes both multi-mapping and unique reads. (E) ChIP-qPCR showing enrichment of MBD7-GFP at the *d35S* promoter driving *LUC* expression at the *YJ* reporter. The data represents the average enrichment from three biological replicates as a percentage of the input and the error bars represent the standard deviation between replicates.

The following figure supplement is available for figure 4:

**Figure supplement 1.** Complementation of the *mbd7-5* phenotype with MBD7-3xHA.

when compared to the MethylC-seq data (*Figure 4C,D*), in that hyper-methylation (primarily in the CG and CHG contexts) was detected at the 3' end of the promoter (*Figure 5A,B*). Also consistent with the MethylC-seq data, the increased methylation observed in the *mbd7-5* mutant was restored to a more wild-type level upon introduction of the *pMBD7::MBD7-3xHA* transgene (*Figure 5A,B*). These findings demonstrate that the two methods of bisulfite sequencing gave similar results. However, unlike the MethylC-seq data (*Figure 1—figure supplement 1C,E*), the traditional bisulfite sequencing definitively shows that the changes in DNA methylation observed in the *lil* and *mbd7* mutants occur at the *35S* promoter driving *LUC* expression. Furthermore, the traditional bisulfite sequencing offers the added benefit of determining whether the average percent methylation across the *35S* promoter represents a uniform distribution of methylation or a bimodal distribution (i.e., some promoters showing high methylation levels and others showing low methylation levels). These analyses revealed a uniform distribution of methylation at the *YJ* reporter (*Figure 5—figure supplement 1*). Thus, it does not appear that there is a significant population of fully unmethylated (or even specifically 3' unmethylated) *35S* promoters that give rise to the observed *LUC* expression.

We next determined the DNA methylation patterns and *LUC* expression levels at the *YJ* reporter in mutants affecting de novo DNA methylation (*ago4* and *nrpe1*) or DNA demethylation (the triple demethylase mutant, *rdd*). These analyses suggest that the hyper-methylation at the 3' end of the *d35S* promoter and *LUC* silencing phenotypes can be genetically uncoupled. At the level of DNA methylation, decreases in non-CG methylation were observed in the *ago4-6* and *nrpe1-11* mutants predominantly at regions of the promoter producing large amounts of 24-nt siRNAs, as predicted for components of the RNA-directed DNA methylation pathway (*Figure 5A*; '24-nt siRNA clusters'). Conversely, all three mutants (*ago4-6*, *nrpe1-11* and *rdd*) showed a hyper-methylation phenotype similar to that observed in the *lil* and *mbd7* mutants at the 3' end of the *35S* promoter (*Figure 5A,B* and *Figure 5—figure supplement 1*). These findings are consistent with: (1) previously identified genetic connections between LIL, MBD7 and ROS1 (*Lang et al., 2015*; *Li et al., 2015b*; *Wang et al., 2015*; *Won et al., 2012*), and (2) studies showing that mutants affecting the establishment of DNA methylation (including *ago4* and *nrpe1*) cause down-regulation of *ROS1* (*He et al., 2011*) via a methylation sensor element in the *ROS1* promoter (*Williams et al., 2015*; *Lei et al., 2015*). Notably, although the *ago4-6*, *nrpe1-11* and *rdd* mutants exhibit similar hyper-methylation phenotypes in the CG and CHG contexts at the 3' end of the *35S* promoter, *LUC* expression is much higher in the *ago4* and *nrpe1* mutants than in the *rdd* mutant (*Figure 5C,D*). While compensatory effects on gene expression due to decreased non-CG methylation at other regions of the *d35S* promoter in the *ago4* and *nrpe1* mutants cannot be fully excluded, these findings suggest that the hyper-methylation at the 3' end of the *d35S* promoter region alone is not sufficient to cause gene silencing. Taken

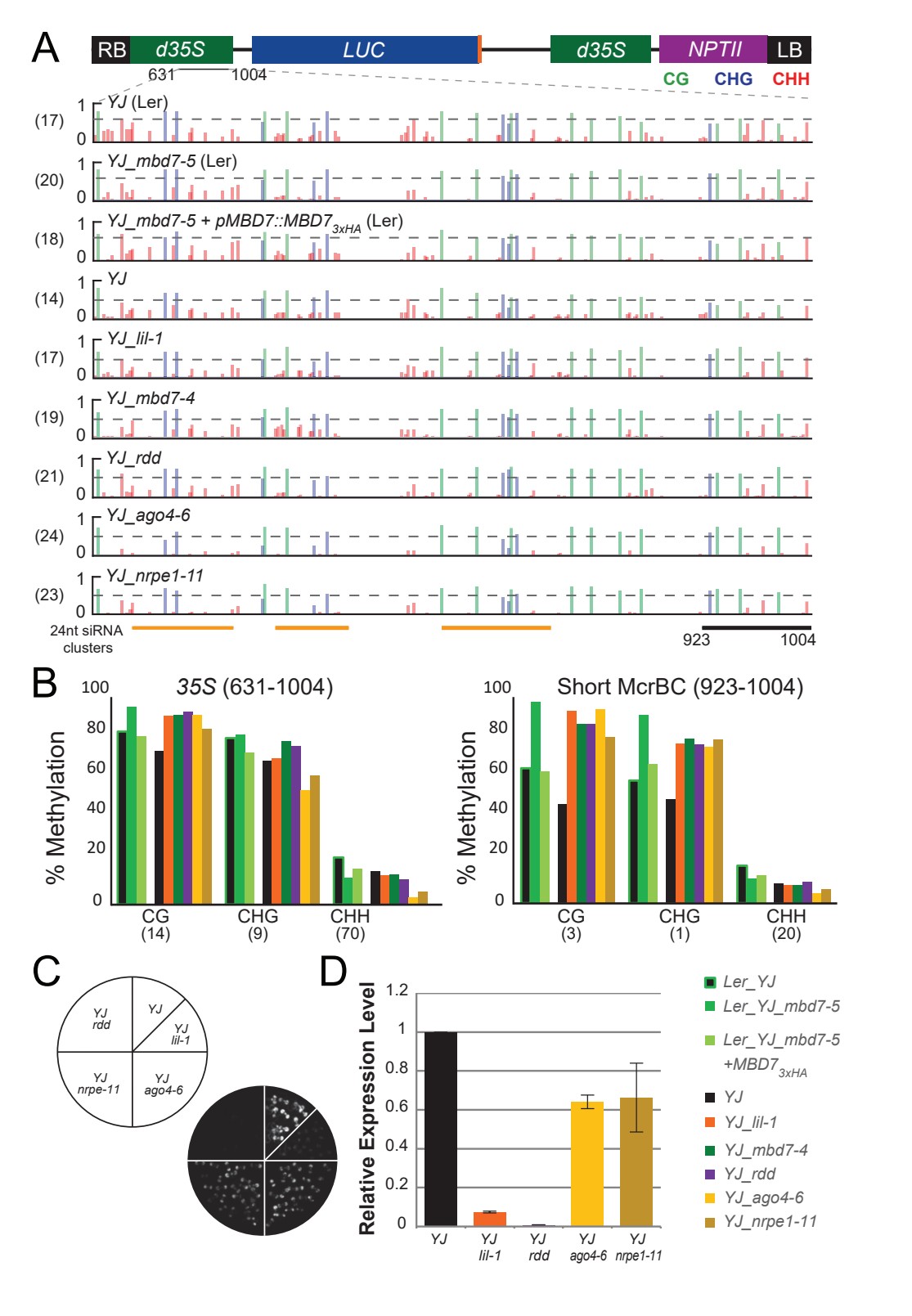

**Figure 5.** Genetic uncoupling of the DNA methylation and *LUC* expression phenotypes at the *YJ* reporter. (**A**) Diagram of the *YJ* transgene indicating the region examined by traditional bisulfite sequencing (631–1004). DNA methylation tracks show the DNA methylation level (0–1, where 1 = 100% methylated) in the CG (green), CHG (blue) and CHH (red) sequence contexts. The tracks represent the average methylation at each position. The
*Figure 5 continued on next page*

*Figure 5 continued*

number of clones per genotype is indicated in parentheses to the left of each track. The orange bars below the methylation tracks denote regions that produce 24-nt siRNA clusters. The dashed horizontal lines spanning the methylation tracks are set relative to the maximal CG methylation at the 3′ end of the *d35S* promoter in the *YJ* reporter to facilitate visual assessment of the changes in DNA methylation in the mutant backgrounds. (**B**) Quantification of the average percent methylation across the second *35S* promoter (631–1004) or a shorter region at the 3′ end of the *35S* promoter (923–1004) within the *YJ* transgene in the lines presented in (**A**). The number of cytosines in each sequence context within the quantified regions is indicated in parentheses. Note that these numbers differ from those presented in *Figure 4* since the traditional bisulfite sequencing only captures methylation on one strand. (**C**) Luciferase luminescence of 10-day-old seedlings as diagrammed on the left. (**D**) Quantification of *LUC* transcript levels by RT-qPCR. *LUC* transcript levels were normalized to *UBIQUITIN5* with the expression level of *LUC* in the *YJ* control set to one. Error bars indicate the standard deviation from two biological replicates.

The following figure supplement is available for figure 5:

**Figure supplement 1.** Traditional bisulfite sequencing at the *YJ* reporter.

together, these analyses reveal that at the *YJ* reporter the MBD7 complex can regulate gene expression in a manner that is largely downstream of DNA methylation.

## Genome-wide profiling shows that *mbd7* and *lil* mutants have minimal effects on DNA methylation

To investigate the role of the MBD7 complex in regulating DNA methylation and to further characterize the relationship between this protein complex and other pathways known to regulate DNA methylation, bisulfite sequencing experiments were conducted. Specifically, the MethylC-seq method (*Urich et al., 2015*) was employed to determine the methylation profiles of paired sets of mutant and control lines at an average of ~30x coverage with ≥97% conversion rates (*Supplementary file 1E and 1F*). Altogether, methylation was profiled in two alleles of *LIL* (*lil-1* and *lil-2* in the *YJ* and *LUCH* backgrounds, respectively), two alleles of *MBD7* (*mbd7-3* and *mbd7-4*, both in the *YJ* background), and the triple DNA demethylase mutant (*rdd* introgressed into the Col background (*Penterman et al., 2007*)). In addition, three biological replicates of *lil-1* (*lil-1*_r1, *lil-1*_r2, and *lil-1*_r5), each with their own controls, were profiled (*Supplementary file 1E*). Using this MethylC-seq data, differentially methylated regions (DMRs) were called using methods very similar to those described in *Stroud et al. (2013)*. In both DMR calling pipelines, an initial set of DMRs were called using the same requirements: an absolute change in methylation of ≥40% for CG, ≥20% for CHG, and ≥10% for CHH with an FDR < 0.01 between the mutant and control samples across 100 bp, non-overlapping regions of the genome. However, the two methods used analogous, though not identical, approaches to account for natural variation in DNA methylation, which is known to occur even amongst siblings of the same ecotype (*Schmitz et al., 2011*; *Becker et al., 2011*). While *Stroud et al. (2013)* sequenced three biological replicates of a wild-type sample and only included DMRS observed between a given mutant and all three wild-type controls, we instead included a control sample for each biological replicate and/or each mutant allele and then only included DMRs conserved between these replicates. Although thousands of DMRs were identified in each of the *lil-1* and *mbd7* datasets, mostly in the CG and CHH contexts (*Supplementary file 2A*), the vast majority of these DMRs were not in common between the different alleles or even between biological replicates (*Supplementary file 2B-F*, see orange highlighted comparisons). Indeed, when the DMRs from the three *lil-1* replicates were compared with the DMRs identified in *lil-2*, *mbd7-3* and *mbd7-4*, only 33 hyper and one hypo DMRs remained (*Figure 6A*, *Supplementary file 2F*; 6-way DMR overlaps, and *Supplementary file 2G-J*), and these regions converged on 20 loci (19 displayed hyper-methylation and one showed hypo-methylation [*Figure 6B* and *Figure 6—figure supplement 1*]). Thus, while numerous DMRs can be identified between any single *mbd7* or *lil* mutant vs. control (*Supplementary file 2A*), very few genomic targets show consistent changes in DNA methylation upon disruption of the MBD7 complex, indicating that these changes may represent natural variation in methylation.

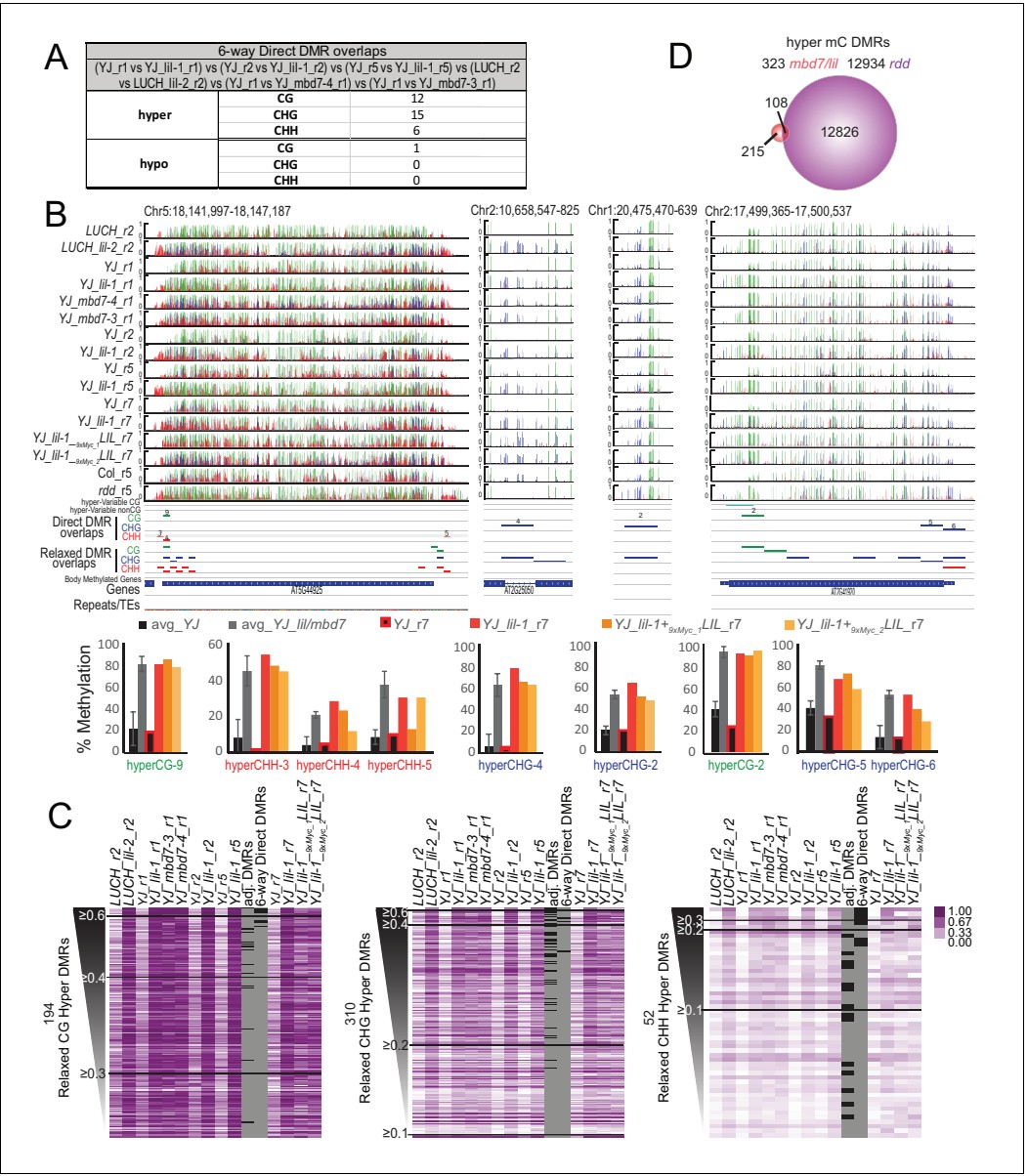

**Figure 6.** Genomic DMRs identified in the *lil* and *mbd7* mutants. (**A**) Table summarizing DMRs that overlap among all the *lil* and *mbd7* datasets '6-way Direct DMR overlaps'. (**B**) Screenshots showing the DNA methylation levels (0–1, where 1 = 100% methylated, with CG, CHG and CHH contexts in green, blue and red, respectively) at several DMRs in the genotypes indicated on the left. Below the DNA methylation tracks are the following additional features: (1) hyper-variable DMRs in the CG and non-CG contexts from *Schmitz et al. (2011)*, (2) hyper DMRs in the CG, CHG and CHH contexts that overlap amongst all the *lil* and *mbd7* datasets ('Direct DMR overlaps'), (3) an expanded set of DMRs identified as detailed in (**C**) and in *Figure 6—figure supplement 2* ('Relaxed DMR overlaps'), (4) body methylated genes as described in *Takuno and Gaut (2012)*, and (5) TAIR10 annotated genes and repeats. The percent methylation over each DMR is quantified in the bar graphs below. The average levels of methylation in controls vs. the *mbd7* and *lil* DMRs are shown in black and grey, respectively. The error bars represent the standard deviation to capture the level of variation between samples. The levels of methylation in the MycLIL complementation data set are also indicted. (**C**) Heatmaps showing the methylation levels at DMRs called using a relaxed set of criteria (e.g., DMRs where all of the samples show hyper-methylation in the CHG context, but some are slightly below the required 20% change cutoff). The hyper DMRs are ranked from most to least robust and changes in methylation ≥0.6, ≥0.4, ≥0.3, ≥0.2 and≥0.1 are indicated. The DMRs indicated in panel A ('6-way_Direct DMRs') and the less stringent DMRs that map to genomic regions adjacent to the direct DMRs ('adj. DMRs') are demarcated by black boxes on the heatmaps. (**D**) Venn diagram showing the hyper mC

*Figure 6 continued on next page*

*Figure 6 continued*

DMRs (see Materials and methods) shared between the *mbd7* and *lil* mutants (red) that overlap with DMRs identified in the *rdd* triple mutant (purple).

The following figure supplements are available for figure 6:

**Figure supplement 1.** Visualization and quantification of DNA methylation at the direct overlap DMRs.

**Figure supplement 2.** Identification of a more relaxed set of DMRs conserved amongst the *mbd7* and *lil* datasets.

**Figure supplement 3.** Complementation of the *lil-1* phenotype with 9xMyc-LIL.

**Figure supplement 4.** Gene expression at previously characterized loci in *mbd7* and *lil* mutants.

Acknowledging that requiring a direct overlap in DMRs between all six samples is quite stringent, a more relaxed set of DMRs was generated to determine whether a larger number of genomic targets dependent on the MBD7 complex would emerge. For these analyses, the methylation levels across the totality of DMRs identified in any of the various *lil* and *mbd7* mutants were first determined (*Supplementary file 2K-P*; MasterDMRLists). DMRs that behaved similarly in all samples were then identified via a clustering analysis and filtered to remove body-methylated genes, which as a group are known to display a higher degree of natural variation in DNA methylation levels (*Schmitz et al., 2011*) (*Supplementary file 2Q-U*; Relaxed_DMR_lists and *Figure 6—figure supplement 2*). This yielded a set of 194 CG hyper DMRs, 310 CHG hyper DMRs and 52 CHH hyper DMRs (*Figure 6C* and *Figure 6—figure supplement 2*). Notably, many of these hyper DMRs are located adjacent to the original set of 33 high stringency DMRs (*Figure 6C*; high stringency DMRs from *Figure 6A* and DMRs located adjacent to these regions are indicated with black bars in the heat-maps). This demonstrates that both methodologies are converging on a similar, small set of genomic loci that are hyper-methylated in both *mbd7* and *lil* mutants. When changes in methylation regardless of the sequence context were combined, these DMRs converged on 323 genomic regions that are distributed across the five chromosome arms (*Figure 6—figure supplement 2D* and *Supplementary file 2V and 2W*; hyper_mC_merged). Notably, only a third of these regions overlap with regions hyper-methylated in the *rdd* mutant background, representing a tiny fraction of the total *rdd* targets (*Figure 6D*). Thus, although there are clear genetic connections between MBD7, LIL and ROS1 in the regulation of several transgenic reporters (*Lang et al., 2015*; *Li et al., 2015b*; *Wang et al., 2015*), as a general rule, the demethylation pathway does not appear to function in a manner that depends solely on either MBD7 or LIL.

To further investigate the functional relevance of the *mbd7* and *lil* hyper DMRs, we determined whether the observed increases in DNA methylation could be complemented by re-introduction of a functional LIL protein. Using stable transgenic plant lines expressing an N-terminal Myc-tagged LIL protein under the control of its endogenous promoter (*pLIL::9xMyc-LIL*), complementation was first confirmed by assessing the *LUC* expression and DNA methylation phenotypes at the *YJ* reporter (*Figure 6—figure supplement 3*). Then, the global patterns of DNA methylation in the *YJ_r7* and *YJ lil-1_r7* parental lines, as well as two sibling complementing lines (*YJ_lil-1+ $_{9xMyc\_1}$LIL* and *YJ_lil-1+ $_{9xMyc\_2}$LIL*), were determined using the MethylC-seq method (*Supplementary file 1E*). In general, the complementing lines more closely resemble the *YJ lil-1_r7* mutant line than the *YJ_r7* control (*Figure 6C*), suggesting minimal complementation overall. Although the possibility of partial complementation at some loci cannot be excluded, even at the most robust set of DMRs, only 1 of 33 DMRs returned to control levels of DNA methylation (*Figure 6—figure supplement 1*, hyper-CHH site 1. Of the remaining sites, eight showed consistent but modest decreases in DNA methylation (indicated by a single asterisks in *Figure 6—figure supplement 1*). However, 5 of these directly overlapped with known hyper-variable regions (*Schmitz et al., 2011*). These findings demonstrate that re-introduction of LIL is not able to efficiently correct the observed hyper-methylation defects, which further supports the notion that these changes represent variation in methylation rather than a direct effect of the *lil* mutation.

## Transcriptome profiling of *mbd7* and *lil* mutants

To further investigate the role of the MBD7 complex in gene regulation, we assessed the effects of the *lil* and *mbd7* mutants on gene expression by transcriptome profiling. Similar to the approach taken for the DNA methylation profiling, two alleles of *LIL* (*lil-1* and *lil-2* in the *YJ* and *LUCH* backgrounds, respectively) and two alleles of *MBD7* (*mbd7-3* and *mbd7-4*) were compared to their corresponding controls (*Supplementary file 3*). Among these mutants, no overlapping mis-regulated genes were identified between the *lil* and *mbd7* alleles (*Supplementary file 3*) and only one of the genes previously identified as a target of MBD7 complex (*i.e.* down-regulated genes associated with hyper DMRs in the Columbia ecotype (*Lang et al., 2015*; *Li et al., 2015b*; *Qian et al., 2014*; *Duan et al., 2017*; *Qian et al., 2012*)) was consistently down-regulated in any of the mutants tested (*Figure 6—figure supplement 4*). As some common alleles of *mbd7* were used in these studies and similar developmental stages were utilized, the factors leading to the differing results remain unclear. However, perhaps differences in growth conditions and/or the differing sensitivities and normalization procedures of the assays used to assess gene expression levels (*i.e.* mRNA-seq vs qPCR), as well as the already low expression levels of the down-regulated genes under our conditions, represent contributing factors (*Figure 6—figure supplement 4*).

Given the absence of detectable endogenous targets transcriptionally regulated by the MBD7 complex, it is likely that this complex functions redundantly with other MBD-ACD complexes and/or is required only under specific conditions. Nonetheless, our findings demonstrate that this complex has the ability to regulate gene expression in a manner largely downstream of DNA methylation at *LUC* reporters. Firstly, we found that loss of the MBD7-LIL complex results in the silencing of these reporter constructs with minimal changes in DNA methylation. Secondly, we demonstrated that hyper-methylation at the 3′ end of the *d35S* promoter in the *YJ* reporter does not appear to be sufficient to cause gene silencing (*Figure 5*). As such, the MBD7 complex joins a small number of factors including MOM1, MORC1, MORC6, ATXR5, and ATXR6, that can act primarily downstream of DNA methylation. However, unlike these other downstream effectors, which function to reinforce gene silencing, MBD7 and LIL represent the first anti-silencing complex placed downstream of DNA methylation, functioning to enable gene expression despite high levels of promoter DNA methylation.

## Discussion

In this study, we identified two MBD-containing protein complexes and demonstrated a role for the MBD7 complex in promoting the expression of methylated transgenes. Furthermore, we found that while mutations in components of the MBD7 complex led to increased DNA methylation at the *d35S* promoter driving *LUC* expression, this methylation alone did not appear to be sufficient to cause *LUC* silencing. These findings, along with genome-wide analyses showing that only a small number of loci displayed a consistent hyper-methylation phenotype in *mbd7* and *lil* mutants, support an alternative hypothesis regarding the role of the MBD7 complex. Rather than functioning as part of the DNA demethylation pathway, as has been previously hypothesized (*Lang et al., 2015*; *Li et al., 2015b*; *Wang et al., 2015*), our results suggest that the MBD7 complex may function in a manner largely downstream of DNA methylation. As very few proteins have been characterized that function downstream of DNA methylation, further characterization of this complex offers the potential to gain much needed insight into the mechanisms by which DNA methylation affects gene expression. Below, we summarize the current knowledge regarding the function and composition of MBD complexes from this and several previous studies (*Lang et al., 2015*; *Li et al., 2015b*; *Wang et al., 2015*; *Zhao et al., 2014*; *Qian et al., 2014*, *2012*; *Li et al., 2012*). In addition, we highlight what is known about other factors that influence gene regulation downstream of DNA methylation.

### The MBD7 complex and the DNA demethylation machinery

The first anti-silencing factors discovered in *Arabidopsis* were a family of DNA glycosylases that recognize and remove methylated cytosine bases, leading to the release of gene silencing in a locus-specific manner (*Zhu, 2009*). Recently, additional anti-silencing factors have been identified through genetic screens and biochemical approaches revealing unanticipated connections between proteins that bind methylated DNA (MBD7 [*Lang et al., 2015*; *Li et al., 2015b*; *Wang et al., 2015*] and ROS4/IDM1 [*Qian et al., 2012*; *Li et al., 2012*]) and proteins with alpha crystallin domains (ACDs)

(LIL/IDL1/IDM3 [*Lang et al., 2015*; *Li et al., 2015b*] and ROS5/IDM2 [*Wang et al., 2015*; *Zhao et al., 2014*; *Qian et al., 2014*]) in regulating the expression of methylated genes. As part of these screening efforts, the methylation status of one endogenous locus (*At1g26400*) and four reporter transgenes have been characterized. In all these cases, similar changes in DNA methylation and gene expression were observed in mutants of the demethylation machinery (either *ros1* single, or *rdd* triple mutants) or the MBD7 complex (*mbd7*, *lil*, *ros4*, and *ros5* mutants). Given this consistent genetic connection, previous studies have concluded that the various MBD and ACD proteins function in a common demethylation pathway with ROS1 to prevent hyper-methylation and enable gene expression, despite the fact that only modest increases in DNA methylation were observed (*Lang et al., 2015*; *Li et al., 2015b*; *Wang et al., 2015*; *Zhao et al., 2014*; *Qian et al., 2014*, *2012*; *Li et al., 2012*). Here we show that the observed hyper-methylation and gene expression defects can be genetically uncoupled at the *YJ* reporter since *LUC* expression levels remain high in *ago4* and *nrpe1* mutants despite hyper-methylation patterns that are identical to those observed in *mbd7*, *lil*, or *rdd* mutant backgrounds. Furthermore, we show that on a genome-wide scale, the genetic connections between MBD7, LIL, and the demethylase machinery are no longer observed. Not only are there very few loci that show a consistent hyper-methylation phenotype across the various *mbd7* and *lil* datasets, but these loci represent a tiny fraction of the hyper-methylated loci in *rdd* mutants. Thus, while we cannot fully rule out a role for the MBD7 complex as a highly locus-specific regulator of the DNA demethylation pathway, nor do we know the extent to which other MBD-ACD complexes might function redundantly with the MBD7 complex and mask its role at endogenous targets and/or connections with the demethylase machinery, we currently favor a model based on our extensive transgene analysis in which the MBD7 complex functions largely downstream of DNA methylation to promote gene expression through a yet unknown mechanism.

Further support for the notion that the primary function of the MBD7 complex is downstream of DNA methylation comes from a comparison of the reporter transgenes used to identify components of this complex. While all these reporters contained *35S* promoters driving the expression of genes that are repressed in mutants of the MBD7 complex, no commonalities in the hyper-methylated regions were identified. For the *YJ* and *LUCH* reporters, hyper-methylation is restricted to the 3′ end of the *d35S* promoter driving *LUC* expression (*Figures 4C* and *5A*). Interestingly, this hyper-methylation corresponds to two TGACG motifs previously shown be bound by a tobacco protein in a methylation sensitive manner (*Kanazawa et al., 2007*). However, this same region does not appear to be hyper-methylated in the *d35S* promoters present in the *SUC2* or *NPTII* reporters. At the *SUC* reporter the hyper-methylation is instead observed at the 5′ end of the *35S* promoter (*Lang et al., 2015*) and the hyper-methylation at the *NPTII* reporter is downstream of the *NPTII* gene, in the *NOS* terminator region (*Wang et al., 2015*). Thus, as there is no clear consensus region that is hyper-methylated amongst these reporters, we posit that the hyper-methylation patterns may represent indirect effects caused by disruption of the MBD7 complex rather than locus specific regulation of DNA methylation.

## Composition of MBD and ACD/HSP20-like complexes

In the present study, MBD5 and MBD7 were found to associate with distinct sets of alpha crystallin domain (ACD)-containing proteins. Furthermore, MBD7 and its associated proteins, but not MBD5, were found to suppress the silencing of methylated luciferase (*LUC*) reporter transgenes, suggesting different roles for the MBD5 and MBD7 complexes. In addition to these purifications, each component of the MBD7 complex (MBD7, ROS4/IDM1, LIL/IDL1/IDM3, and ROS5/IDM2) has now been individually affinity purified and the co-purifying factors and their approximate relative abundances have been determined using Mass Spectrometry (*Figure 2—figure supplement 1* and refs [*Lang et al., 2015*; *Li et al., 2015b*]). With the exception of our 3xHA tagged MBD7 purification, all these Mass Spectrometry experiments yielded peptides corresponding to all four factors. However, in *Lang et al. (2015)* the purification of MBD7 yielded the most peptides matching LIL/IDM3, with relatively fewer hits to IDM1 and ROS5/IDM2, suggesting there may be a stable sub-complex of MBD7 and LIL/IDM3 in addition to the larger MBD7 complex. Taken together, these data demonstrate the existence of multiple MBD complexes, expanding the number of known MBD-ACD complexes, and revealing a functional difference between MBD5 and MBD7 complexes.

As a complement to the affinity purifications, many of the pair-wise interactions between the various MBD and ACD proteins have been determined using Y2H assays (*Lang et al., 2015*; *Li et al.,*

*2015b*; *Wang et al., 2015*; *Zhao et al., 2014*; *Qian et al., 2014*). In several cases, subdomains important for these interactions have been mapped, revealing additional insights into the organization of these protein complexes. For example, the two ACD domain-containing proteins (LIL/IDL1/ IDM3, and ROS5/IDM2) interact directly with each other (*Li et al., 2015b*), with MBD7 (*Lang et al., 2015*; *Li et al., 2015b*; *Wang et al., 2015*) and with ROS4/IDM1(*Li et al., 2015b*; *Zhao et al., 2014*; *Qian et al., 2014*), whereas MBD7 and ROS4/IDM1 fail to show a direct interaction (*Lang et al., 2015*; *Li et al., 2015b*). This suggests the ACD-domain proteins may function to bridge the association between MBD7 and ROS4/IDM1. Furthermore, the interaction domains between ROS4/IDM1 and either LIL/IDL1/IDM3 or ROS5/IDM2 map to different regions of the ROS4/IDM1 protein (*Li et al., 2015b*; *Zhao et al., 2014*) suggesting that both these ACD domain proteins could interact with ROS4/IDM1 at the same time. Similarly, Lang et al. and this study show that two adjacent regions in MBD7 are each sufficient to interact with LIL/IDL1/IDM3 (*Figure 2—figure supplement 1*). Thus, one can envision scenarios in which the two ACDs could compete for binding to the interaction domains in ROS4/IDM1 or MBD7, or cooperatively bind given their affinities for each other. Beyond these pair-wise interactions, *Zhao et al. (2014)* demonstrated that ROS5/IDM2 forms higher-order structures that migrate at ~670 kDa after gel filtration, consistent with a 16-mer complex. This finding demonstrates that at least ROS5/IDM2, and possibly other ACD proteins, have the ability to oligomerize in a manner similar to canonical HSP20 proteins (*Scharf et al., 2001*). Although it remains unclear which interactions and structural features are required for the function of the MBD7 complex, the fact that loss of any component abolishes activity suggests a close tie between structure and function.

## Regulation of gene expression downstream of DNA methylation

While our understanding of the mechanisms through which gene expression is regulated downstream of DNA methylation remains quite limited, several factors have been shown to release gene silencing while having minimal effects on DNA methylation. Many of these factors, including MORC1, MORC6, ATXR5 and ATXR6, influence chromatin organization on a gross scale, such that loss of these factors results in de-compaction of peri-centromeric heterochromatin (i.e., chromocenters) (*Moissiard et al., 2012*; *Jacob et al., 2009*) and the selective up-regulation of genes and transposons located within these regions (*Moissiard et al., 2012*; *Jacob et al., 2009*). On the other hand, MOM1 appears to act on a finer scale to influence gene expression without significantly perturbing chromatin compaction (*Mittelsten Scheid et al., 2002*; *Probst et al., 2003*). In *mbd7* and *lil* mutants, no obvious defects in chromocenter formation were observed, suggesting the MBD7 complex may act at a local level to enable the expression of methylated genes, perhaps antagonizing the function of MOM1, which has been shown to regulate expression at the *LUCH* reporter (*Won et al., 2012*). Alternatively, since there are several other MBD proteins known to bind methylated DNA (e.g., MBD5 and MBD6 (*Zemach and Grafi, 2003*; *Scebba et al., 2003*; *Ito et al., 2003*; *Zemach et al., 2005*)) and several other ACD proteins closely related to LIL/IDL1/IDM3 and ROS5/ IDM2 (*Scharf et al., 2001*) (*Figure 1—figure supplement 3*), these factors may act redundantly to regulate chromatin structure on a more global level.

Moving forward, it will be important to continue exploring the composition, dynamics, and regulatory roles of the various MBD and ACD complexes, and to begin investigating the genetic relationships between these complexes. Such experiments have the potential to uncover genomic targets regulated by these complexes in a redundant manner, and perhaps identify specific roles for these complexes during development. Finally, it will also be important to assess the relative contributions of the enzymatic and structural features associated with these complexes in modulating the local chromatin environment. This will provide further mechanistic insights into how different MBD and ACD complexes function together to regulate the expression of methylated regions of the genome.

## Materials and methods

### Plant materials and transgenic lines

*Arabidopsis thaliana* Columbia-0 (Col) and Landsberg *erecta* (L*er*) ecotype plants were used in the present study. The *LUC*-based reporters *LUCH* (*Won et al., 2012*) and *YJ* (*Li et al., 2016*) are in the *rdr6-11* mutant background (*Peragine et al., 2004*) in all lines used for this study and were originally

generated in the Col background. The *lil-1*, *ros4-2* and *ros4-3* mutants were recovered from an EMS screen in the *YJ* background while the *lil-2* mutant was recovered from a T-DNA screen in the *LUCH* background. The following additional mutants were introduced into the *YJ* background (Col ecotype) by crossing: *mbd7-3* (GABI_067_A09) (*Kleinboelting et al., 2012*) (formerly named *mbd7-1* in (*Li et al., 2015b*) and renamed to avoid duplicate allele names with other publications (*Lang et al., 2015*)), *mbd7-4* (SALKseq_080774) (unpublished), *mbd5-3* (SAILseq_750_A09) (unpublished), *ros5-4* (SALK_138229 also kown as *idm2-2* (*Qian et al., 2014*; *Alonso et al., 2003*), *ago4-6* (*Alonso et al., 2003*), *nrpe1-11* (*Pontier et al., 2005*), and *ros1-3 dml2-1 dml3-1* (*Penterman et al., 2007*). Finally, *mbd5-1* (CSHL_ET8226) (*Sundaresan et al., 1995*) and *mbd7-5* (GT_5_107435) (*Sundaresan et al., 1995*), which are in the L*er* ecotype, were crossed into the *YJ* reporter that was introgressed into the L*er* ecotype through 5 backcrosses.

To generate the *MBD5-3xFlag*, *MBD7-3xHA*, and *MBD7-3xFlag* transgenes, the genomic regions of *MBD5* and *MBD7*, including their endogenous promoter regions, were amplified by PCR using the primer pairs JP3845/JP3891 and JP5523/JP5525, respectively (*Supplementary file 4*). The PCR products were cloned into the pENTR/D/TOPO vector per manufacturer's instructions (Invitrogen). A carboxy terminal 3xFlag or 3xHA tag (described in [*Law et al., 2011*]) was added at an AscI site present in the pENTR/D/TOPO backbone downstream of the *MBD5* or *MBD7* inserts. The resulting plasmids were recombined into a modified version of the pEG302 plasmid as described in *Johnson et al. (2008)* using a Gateway LR clonase kit (Invitrogen) and transformed into Col or *YJ mbd7-5* plants using the floral dip method.

To generate the *pLIL:9xMyc-LIL* transgene, the genomic region of *LIL* encompassing the promoter and coding sequence up to the stop codon was amplified by PCR using primers HSP20-proF1 and HSP20-R3 from Col genomic DNA. The genomic fragment was cloned into pENTR/D-TOPO (Invitrogen, K2400-20) to result in pENTR/D-TOPO-LIL. Site-directed mutagenesis was performed using the Stratagene kit XL II with this clone using primers HSP20-mut3 and HSP20-mut4 to introduce a *Kpn*I site near the start codon of LIL. A 9xMyc-BLRP fragment was released from the pCR2.1::KpnI 9×Myc-BLRP plasmid (*Law et al., 2010b*) by *Kpn*I digestion and cloned into the *Kpn*I site of the pENTR/D-TOPO-LIL plasmid to result in an N-terminal fusion of *9xMyc* to *LIL*. The insert in the entry vector was then recombined into pGWB204 (*Nakagawa et al., 2007*) using a Gateway LR Clonase kit (Invitrogen, 11791–019) and transformed into *YJ lil-1* plants using the floral dip method.

### *LUCH* and *YJ* genetic screens

Mutant populations in the *YJ* and *LUCH* backgrounds were generated via ethyl methanesulfonate (EMS) or T-DNA mutagenesis, respectively. For EMS mutagenesis, 1 ml of seeds (around 10,000 seeds) were washed with 0.1% Tween 20 for 15 min, treated with 0.2% EMS for 12 hr and washed three times with 10 ml water for 1 hr with gentle agitation. For T-DNA mutagenesis, pEarleygate303 was modified to remove the Gateway cassette then transformed into the *LUCH* line. Mutants with lower LUC activity, based on LUC live imaging, were isolated in the M2 generation of the EMS population and the T2 generation of the T-DNA population. The isolated mutants were backcrossed to the respective parental lines two times before further analysis.

### Mapping of the *ros4-2* and *ros4-3* mutations

The *YJ ros4-2* and *YJ ros4-3* mutants were isolated from the *YJ* EMS screen and crossed to the corresponding *YJ* lines in the L*er* background to generate the F2 mapping populations. For *YJ ros4-2*, 28 F2 plants with reduced LUC activity were used for rough mapping, and the mutation was linked to the center of the upper arm of chromosome 3. SSLP and dCAPS markers were designed using identified polymorphisms between the Col and L*er* accessions (http://arabidopsis.org/browse/Cereon/index.jsp). Fine mapping narrowed the region to a 280 kb window spanning the K15M2, F4B12, K7L4, MJK13, MQD17 and MSJ11 BAC clones. Candidate gene sequencing uncovered a G-to-A mutation that introduced a premature stop codon in the seventh exon of *At3g14980*, and the mutation was subsequently referred to as *ros4-2*. For *YJ ros4-3*, 27 F2 plants with reduced LUC activity were used for rough mapping, and linkage to the same mapping region of *ros4-2* was observed. Sequencing of *At3g14980* revealed a G-to-A mutation that introduced a premature stop codon in the second exon.

## Mapping of the *lil-1* and *lil-2* mutations

The *YJ lil-1* and *LUCH lil-2* mutants were isolated from the *YJ* EMS screen and the *LUCH* T-DNA screen, respectively, and crossed to the corresponding *LUC* lines in the L*er* background to generate the F2 mapping populations. For *YJ lil-1*, 32 F2 plants with reduced LUC activity were used for rough mapping, and the mutation was linked to the center of the upper arm of chromosome 1. SSLP and dCAPS markers were designed using identified polymorphisms between the Col and L*er* accessions (http://arabidopsis.org/browse/Cereon/index.jsp). Fine mapping narrowed the region to a 160 kb window spanning the F5M15, F2D10 and F9H16 BAC clones. Candidate gene sequencing uncovered a G-to-A mutation in the splice acceptor site of *At1g20870*, and the mutation was subsequently referred to as *lil-1*. For *LUCH lil-2*, 27 F2 plants with reduced LUC activity were used for rough mapping, and linkage to the same mapping region of *lil-1* was observed. Sequencing of *At1g20870* revealed a C-to-T mutation that introduced a premature stop codon in the first exon.

## Yeast two-hybrid screen

The full-length coding sequence of *LIL* was amplified with primers HSP20T7fl F and HSP20T7fl R and cloned into the bait vector pGBKT7 at the *Nde*I and *Bam*HI sites to be fused in-frame with the sequence encoding the GAL4 DNA-binding domain (BD). The *Arabidopsis* cDNA library cloned into the prey vector pGADT7-RecAB was constructed by Clontech. All experiments using the yeast two-hybrid system were carried out according to the manufacturer's instructions (Clontech, Matchmaker GAL4 Two-Hybrid System 3 and Libraries User Manual, PT3247-1). The bait plasmid pGBKT7-LIL and the prey library DNA were co-transformed into the yeast strain AH109. The resulting progeny were first selected on SD/-Leu/-Trp/-His/-Ade plates then tested for β-galactosidase activity to eliminate false positives. Plasmids harboring positive prey cDNAs were isolated and sequenced to ensure that the cDNAs had been fused in-frame with the sequence encoding the GAL4 AD domain.

To identify the interaction domains, the alpha crystallin/Hsp20 domain of *LIL* was amplified with primers HSP20T7D1F and HSP20T7D1R and cloned into pGBKT7 as the bait LILD1. The *LIL* sequence 5′ to the alpha crystallin/Hsp20 domain was amplified with primers HSP20T7N3F and HSP20T7N3R and cloned into pGBKT7 as the bait LILN3. Full-length or truncated *MBD5*, *MBD6* and *MBD7* cDNAs were cloned into pGADT7. Full-length *MBD5* coding sequence was amplified with primers MBD5ADfl F and MBD5ADfl R; Full-length *MBD6* coding sequence was amplified with primers MBD6ADfl F and MBD6ADfl R; Full-length *MBD7* coding sequence was amplified with MBD7ADfl F and MBD7ADfl R. The methyl-CpG-binding domain of *MBD5* was amplified with primers EcoRI-MBD5d and MBD5d-BamHI and cloned into pGADT7 as prey MBD5d. Similarly, the methyl-CpG-binding domain of *MBD6* was amplified with MBD6d-BamHI and EcoRI-MBD6d and cloned into pGADT7 as prey MBD6d. For MBD7, the second methyl-CpG-binding domain, including partial sequence of the third methyl-CpG-binding domain, was amplified with EcoRI-MBD7d2 and MBD7d2-BamHI and cloned into pGADT7 as prey MBD7d2. The third methyl-CpG-binding domain of MBD7 was amplified with EcoRI-MBD7d3 and MBD7d3-BamHI and cloned into pGADT7 as prey MBD7d3. Sequences of primers used in the plasmid construction can be found in *Supplementary file 4*. Colonies containing both bait and prey plasmids were selected by growing yeast on selective dropout medium lacking Trp and Leu (SD/-Trp/-Leu) at 30°C. They were subsequently plated on selective dropout medium lacking Trp, Leu, Ade and His (SD/-Trp/-Leu/-Ade/-His) and grown at 30°C to test interactions between bait and prey.

## Affinity purification and Mass Spectrometry

Affinity purification and Mass Spectrometry of MBD5 and MBD7 were performed largely as described in *Law et al. (2010b)*. Briefly, for each replicate of MBD5, approximately 14 g of flower tissue from MBD5-3xFlag transgenic T$_4$ plants, or from wild type Col plants, were ground in liquid nitrogen. For replicates one and two, the tissue was re-suspended in 75 mL of lysis buffer 1 (LB1: 50 mM Tris pH7.6, 150 mM NaCl, 5 mM MgCl$_2$, 10% glycerol, 0.1% NP-40, 0.5 mM DTT, 1 µg/µL pepstatin, 1 mM PMSF and one protease inhibitor cocktail tablet (Roche, 14696200)), while replicate three was re-suspended in 75 mL of low salt lysis buffer 2 (LB2: LB1 replacing 150 mM NaCl with 100 mM NaCl). In all three cases, immunoprecipitation utilized 250 µL of 50% M2 FLAG-agarose slurry (Sigma, A2220) and proteins were eluted from the beads by competition with 3xFLAG peptide (Sigma, F4799).

For replicates one and two of MBD7, approximately 10 or 15 g of flower tissue from MBD7-3xHA transgenic $T_4$ plants or from wild type Col plants, respectively, were ground in liquid nitrogen and re-suspended in either 50 mL of LB1 (replicate 1) or 75 mL of a triton lysis buffer 3 (LB3: LB1 replacing 0.1% NP-40 with 1% Triton X-100) (replicate 2). Immunoprecipitation utilized 250 µL of 50% HA-conjugated slurry (Roche, 11815016001) and proteins were eluted from the beads by competition with HA peptide (Thermo, 26184). For the third replicate of MBD7, approximately 10 g of flower tissue from MBD7-3xFLAG transgenic plants or from wild type Col plants were ground in liquid nitrogen and re-suspended in 50 mL of lysis buffer 1 (LB1). Immunoprecipitation utilized 250 µL of 50% Anti-FLAG M2 Magnetic bead slurry (Sigma, M8823) and proteins were eluted from the beads by competition with 3xFlag peptide (Sigma, F4799). Eluted proteins were TCA precipitated and subjected to Mass Spectrometry as described in *Law et al. (2010b)*.

## Luciferase imaging

For *Figure 1*, *Figure 3* and *Figure 1—figure supplement 1*, seeds were surface-sterilized and planted on half-strength Murashige and Skoog (MS) media supplemented with 0.8% agar and 1% sucrose then stratified at 4°C for three days. Plants were grown in a growth incubator at 23°C under continuous light. 10-day-old seedlings were used for all of the experiments. Luciferase live imaging was performed as previously described (*Won et al., 2012*).

For *Figure 2* (YJ lines), *Figure 5*, *Figure 2—figure supplement 3*, *Figure 4—figure supplement 1*, and *Figure 6—figure supplement 3*, seeds were surface-sterilized and planted on Linsmaier and Skoog (LS) media (Caissen, LSP03) supplemented with 0.8% agar, then stratified at 4°C for three days. Plants were grown in a growth incubator at 23°C under short day conditions (8 hr light, 16 hr dark). 10-day-old seedlings were used for all of the experiments. Luciferase live imaging was performed using a CCD camera (Andor, iKon-M 934) and PlantLab software (BioImaging Solutions).

## 5-aza-2'-deoxycytidine treatment and Luciferase imaging

For 5-Aza-dC (Sigma, A3656) treatment, plants were grown on Murashige and Skoog (MS) media containing 0.8% agar, 1% sucrose and 7 µg/ml 5-Aza-dC for 2 weeks. Luciferase live imaging was performed as previously described (*Won et al., 2012*).

## RNA extraction and RT-PCR

For *Figures 1* and *2* (*LUCH* lines), *Figure 3* and *Figure 1—figure supplement 1*, RNA was extracted from a pool of 10-day-old seedlings using TRI reagent (Molecular Research Center, TR118) and treated with DNaseI (Roche, 04716728001). cDNA was synthesized using RevertAid Reverse Transcriptase (Thermo Scientific, EP0441) and oligo-dT primer (Thermo Scientific, SO131). RT-qPCR was performed on a Bio-Rad C1000 thermal cycler equipped with a CFX detection module using iQ SYBR Green Supermix (Bio-Rad, 170–0082). For *Figure 2* (YJ lines), *Figure 5*, *Figure 2—figure supplement 3*, *Figure 4—figure supplement 1*, and *Figure 6—figure supplement 3*, RNA was extracted from a pool of 10-day-old seedlings using Zymo Research Quick-RNA Miniprep Kit (Zymo, R1054S). cDNA was synthesized using Applied Biosystems High Capacity cDNA Reverse Transcription Kit (Applied Biosystems, 4368814). RT-qPCR was performed on a Bio-Rad CFX384 Real-Time System using iTaq Universal SYBR Green Supermix (Bio-Rad, 172–5124). Quantification of transgene expression was performed in biological triplicate, unless otherwise indicated. All experiments were conducted per the manufacturers' instructions. The primers used in the study are listed in *Supplementary file 4*.

## McrBC-PCR

Genomic DNA was extracted from a pool of 10-day-old seedlings using the CTAB method (*Rogers and Bendich, 1985*). Three units of McrBC (New England Biolabs, M0272) were used to treat 200–500 ng of DNA at 37°C for 25 min to overnight. A mock experiment was performed in parallel using DNA that had not been treated with McrBC. Either regular PCR or quantitative PCR using iTaq Universal SYBR Green Supermix (Bio-Rad, 172–5124) was performed to determine the level of methylation. For both regular PCR and qPCR, *ACTIN1*, which lacks cytosine methylation, was used as the internal loading control. The relative methylation was calculated as $100-100 \times 2^{Ct(mock) - Ct}$

(treated), such that higher relative levels of PCR amplification correspond to higher levels of methylation. The primers used in the study are listed in *Supplementary file 4*.

## Western blotting

Western blot analysis of MBD7-3xHA transgenic lines was performed using 0.1 g of flower tissue. Tissue was mechanically disrupted and homogenized in cold IP buffer (50 mM Tris, pH 7.6, 150 mM NaCl, 5 mM MgCl$_2$, 10% Glycerol, 0.1% NP40) with protease inhibitors. The lysate was resolved on a 10% Bis-Tris Criterion XT Gel (Bio-Rad, 345–0112) then transferred to a PVDF membrane (GE Healthcare Bio-Sciences, 10600023) and probed with anti-HA-Peroxidase (Roche, 12013819001) (1:2000). MBD7-3xHA was detected by autoradiography using ECL2 Western Blotting Substrate (Pierce, 80196).

Western blot analysis of 9xMyc-LIL transgenic lines was performed using 0.3 g of 10-day-old seedlings. Tissue was mechanically disrupted and homogenized in cold IP buffer (50 mM Tris, pH 7.6, 150 mM NaCl, 5 mM MgCl$_2$, 10% Glycerol, 0.1% NP40) with protease inhibitors. 9xMyc-LIL was immunoprecipitated using Dynabeads Protein G (Invitrogen, 10004D) incubated with monoclonal anti-Myc Tag antibody (Millipore, 05–724). The immunoprecipitate was resolved on a 10% TGX Mini-Protean Gel (Bio-Rad, 456–8035), transferred to a PVDF membrane (GE Healthcare Bio-Sciences, 10600023) and probed with a monoclonal anti-Myc primary antibody (Millipore, 05–724) (1:4000) and a HRP-conjugated Goat anti-mouse secondary antibody (BioRad, 170–6516) (1:5000). 9xMyc-LIL was detected by autoradiography using ECL2 Western Blotting Substrate (Pierce, 80196).

## Traditional bisulfite sequencing

Approximately 2 µg of genomic DNA, extracted using the CTAB method from 10-day-old seedlings, was bisulfite-treated using the MethylCode Bisulfite Conversion Kit (Invitrogen, MEV50). Amplification of specific genomic loci was performed using transgene-specific primers (*Supplementary file 4*; d35S2 tBS Forward and d35S2 tBS Reverse), KAPA HiFi HotStart Uracil+ ReadyMix PCR Kit (KAPA Biosystems, KK2801) and the following PCR conditions: 95°C for 5 min; 98°C for 30 s; 2 cycles of 98°C for 30 s, 66.5°C for 1 min and 72°C for 1 min; 2 cycles of 98°C for 30 s, 65.5°C for 1 min and 72°C for 1 min; 2 cycles of 98°C for 30 s, 64.5°C for 1 min and 72°C for 1 min; 2 cycles of 98°C for 30 s, 63.5°C for 1 min; 32 cycles of 98°C for 30 s, 62.5°C for 1 min and 72°C for 1 min; and 72°C for 15 min. PCR products were run on a 2% agarose gel and ~500 bp bands were purified using the QIAGEN Gel Extraction Kit (Qiagen, 28704). Purified PCR products were cloned into the pCRII-TOPO vector using the Invitrogen ZeroBlunt TOPO PCR cloning kit (Invitrogen, 450245). The resulting plasmids were transformed into One-Shot TOP10 Chemically Competent *E. coli* cells (Invitrogen, C404003). Sequencing was performed from bacterial glycerol stocks of 24 different colonies using the M13 Forward (−21) primer. A small number of clonal sequences as well as sequences with regions of potential non-conversion were removed from the analysis. Sequences were aligned to a reference using the CLC Main Workbench (www.clcbio.com) and visualized using cymate. The percent methylation (number of methylated cytosines divided by the total number of cytosines) was calculated for each cytosine. The average percent methylation was reported for specific ranges within the *35S* sequence.

## Small RNA library construction, sequencing, and bioinformatics analysis

Total RNA was size-fractionated by electrophoresis and RNAs 15 to 40 nt in length were purified and subjected to library construction. Small RNA libraries were prepared using the TruSeq Small RNA Sample Preparation Kit (Illumina, RS-200–0012) according to the manufacturer's instructions and sequenced with Illumina's HiSeq2000 platform at the UCR Institute for Integrative Genome Biology (IIGB) genomic core facility. 3' adapter sequences were trimmed from the raw reads using custom Perl scripts (*Source code 1*). Reads <18 nt after adapter trimming or corresponding to rRNA, tRNA, snRNAs and snoRNAs were discarded. The remaining reads were aligned to the TAIR10 Arabidopsis genome or the *YJ/LUCH* transgene sequence using bowtie allowing for perfect match only and multiple mapping (-v 0 –m 1000) or unique mapping (-v 0 –m 1).

## MethylC-seq library construction and sequencing

MethylC-seq libraries corresponding to the '_r2, _r5, and_B' series (*Supplementary file 1E*) were prepared as follows: Genomic DNA was extracted using the DNeasy Plant Mini Kit (Qiagen, 69104). One microgram of genomic DNA was sonicated into fragments 150 to 300 bp in length using a Diagenode Bioruptor, followed by purification with the PureLink PCR Purification Kit (Invitrogen, K3100-01). DNA ends were repaired using the End-It DNA End-Repair Kit (Epicentre, ER0720), and the DNA fragments were purified using the Agencourt AMPure XP-PCR Purification system (Beckman Coulter, A63880). The purified DNAs were adenylated at the 3' end using the polymerase activity of Klenow Fragment (3'→5' exo-) (New England Biolabs, M0212), followed by purification using the Agencourt AMPure XP-PCR Purification system. The methylated adapters in the TruSeq DNA Sample Preparation Kit (Illumina, FC-121–2001) were ligated to the DNA fragments using T4 DNA Ligase (New England Biolabs, M0202). After purification using AMPure XP beads, less than 400 ng DNA was subjected to bisulfite conversion using the MethylCode Bisulfite Conversion Kit (Invitrogen, MECOV-50). PfuTurbo Cx Hotstart DNA polymerase (Agilent, 600414) and the following PCR conditions were used for amplification: 95°C for 2 min; 9 cycles of 98°C for 15 s, 60°C for 30 s and 72°C for 4 min; and 72°C for 10 min.

All the remaining MethylC-seq libraries (*Supplementary file 1E*) were prepared following a slightly modified protocol as detailed in *Urich et al. (2015)*. Genomic DNA was extracted from 10-day-old seedlings using the DNeasy Plant Mini Kit (Qiagen, 69104). 2 µg of genomic DNA was sonicated into 200 bp fragments using a Covaris S2 Sonicator, followed by purification with Sera-Mag Magnetic SpeedBeads (ThermoScientific, 65152105050250). DNA ends were repaired using the End-It DNA End-Repair Kit (Epicentre, ER81050), and the DNA fragments were purified using Sera-Mag Magnetic SpeedBeads. The purified DNAs were adenylated at the 3' end using the polymerase activity of Klenow Fragment (3'→5' exo-) (New England Biolabs, M0212), followed by purification using Sera-Mag Magnetic SpeedBeads. The methylated adapters in the NEXTflex Bisulfite-Seq Barcodes Kit (BIOO Scientific, 511911) were ligated to the DNA fragments using T4 DNA Ligase (New England Biolabs, M0202). After purification using Sera-Mag Magnetic SpeedBeads, the DNA was subjected to bisulfite conversion using the MethylCode Bisulfite Conversion Kit (Invitrogen, MECOV-50). KAPA HiFi HotStart Uracil+ ReadyMix PCR Kit (KAPA Biosystems, KK2801) and the following PCR conditions were used for amplification: 95°C for 2 min; 98°C for 30 s; 4 cycles of 98°C for 15 s, 60°C for 30 s and 72°C for 4 min; and 72°C for 10 min. After purification using Sera-Mag Magnetic SpeedBeads, the libraries were pooled and then sequenced and processed by the Next Generation Sequencing Core at the Salk Institute for Biological Studies.

## Illumina sequencing

MethylC-seq libraries corresponding to the 'YJ_r2, YJ_lil-1_r2 and Col_B' samples (*Supplementary file 1E*) were sequenced solely using HiSeq 2000 with the 101-cycle single-end sequencing mode (Illumina). MethylC-seq libraries corresponding to the '_r1, _r1_r2, _r6, and _r7,' series were sequenced solely using the Illumina HiSeq 2500 v4 at the Salk NGS Core with the 50-cycle single-end sequencing mode (Illumina). MethylC-seq libraries corresponding to the 'Col_r5, rdd_r5, LUCH_r2, LUCH_lil-2_r2, YJ_r5, and YJ_lil-1_r5' samples (*Supplementary file 1E*) where sequenced using both sequencers (HiSeq 2000 and 2500) using the 101- and 50-cycle single-end modes, respectively, and the data was combined for the final analyses.

## MethylC-seq analysis at the *YJ* and *LUCH* transgenes

Illumina sequence reads were filtered to remove duplicate reads either with prinseq (*Schmieder and Edwards, 2011*), using the remove exact duplicates option (-derep 1), or the BSseeker2 (*Guo et al., 2013*) filterReads.py script, using default conditions. The reads were then mapped to the *YJ* or *LUCH* transgene sequences (*Supplementary file 1D*; LUC_transgenes.genome) using bsmap-2.74 (*Xi and Li, 2009*) with the default parameters, allowing two mismatches (-v 2). Since the *d35S* promoters driving the expression of the *LUC* and *NPTII* genes are 94% identical, both unique and multi-mapping reads were included, with the maximum number of equal best hits set to 2 (-w 2). The percent methylation levels (mC reads/total C reads x 100) at each cytosine were quantified using the bsmap methratio.py script, reporting loci with zero methylation ratios (-z). The data was converted to a wiggle format for genome browser visualization with a coverage filter set to 5

(*Supplementary file 1D*; TransgeneWig). In addition, the methylation levels were also determined using only uniquely mapping reads using the BSseeker2 bs_seeker2-align.py script, using the bowtie1 aligner (version bowtie-1.0.0) (*Langmead et al., 2009*) and allowing for two mismatches (-m 2) (*Figure 1—figure supplement 1C*). Mapping and coverage statistics are presented in *Supplementary file 1A and 1B*, respectively.

## Genome-wide MethylC-seq mapping and quantification of DNA methylation levels

Illumina reads were filtered to remove duplicated and low quality reads using the BSseeker2 Filter-Read.py script and then mapped to the TAIR10 genome using the BSseeker2 bs_seeker2-align.py script, using the bowtie1 aligner (version bowtie-1.0.0) (*Langmead et al., 2009*) and allowing for two mismatches (-m 2). Mapping and coverage statistics are presented in *Supplementary file 1E*.

The percent methylation level at each cytosine was calculated using the BSseeker2 bs_seeker2-call_methylation.py script requiring a minimum coverage of 4 reads (-r 4). The resulting CGmap files were used to generate wiggle files containing the percent methylation levels of cytosines covered by at least four reads in the CG, CHG, and CHH contexts individually (*Supplementary file 1F*; Cov4_no-Filter_Wig). The global % methylation in the CG, CHG, and CHH contexts is presented in *Supplementary file 1E*.

## DMR calling and DMR lists

DMRs were identified using parameters outlined in *Stroud et al. (2013)*. Briefly, the genome was split into 100 bp, non-overlapping bins and the methylation level across each bin in the CG, CHG, or CHH context was calculated independently using the percent methylation values in the wiggle files (see **Genome-wide MethylC-seq mapping and quantification of DNA methylation levels**). The methylation values in each bin were then compared between two samples to call DMRs with the following requirements: (1) To account for 100 bp regions of the genome with low numbers of cytosines and for regions that display lower than average coverage, only bins with at least four cytosines in the given context that are covered by at least four reads in both samples being compared were included in the DMR analysis. (2) Only bins with an absolute change in methylation of 0.4, 0.2, and 0.1 for the CG, CHG, and CHH contexts, respectively, and with an adjusted p-value of ≤0.01 were identified as DMRs. The number of DMRs identified for each mutant dataset and their genomic locations are presented in *Supplementary file 2A* and *Supplementary file 1F*; DMRs, respectively.

To determine directly overlapping DMRs between datasets, the bedops interest (-i) (*Neph et al., 2012*) function was utilized to identify common DMRs, and to maintain a common DMR size of 100 bp the bedops chop (-w 100) function was used. The number of DMRs that overlap between 2, 3, 4, 5 or all six datasets and their genomic locations are shown in *Supplementary file 2B-F* and *Supplementary file 2G-J*, respectively. To identify DMRs co-regulated by the MBD7-LIL complex in a more relaxed manner, we first generated six master lists of DMRs (hyper and hypo DMRs in the CG, CHG, or CHH contexts; *Supplementary file 2K-P* Master_DMR_lists; relevant to *Figure 6C* and *Figure 6—figure supplement 2A*) that included all the DMRs called amongst the 6 pairs of samples (3 replicates of *lil-1*, *lil-2*, *mbd7-3* and *mbd7-4* with their respective controls). The methylation levels at the DMRs were then determined (see **Heatmaps**) and clusters of DMRs showing either increased or decreased DNA methylation levels across all the *lil* and *mbd7* datasets were selected (*Figure 6—figure supplement 2A* and *Supplementary file 2Q-U*; Relaxed_DMR_lists). To determine the overlap of these DMRs with previously annotated features including body methylated genes (*Takuno and Gaut, 2012*), the bedops –element-of and –not-element-of functions were used with the overlap threshold set at 1 bp. Overlaps with body-methylated genes were also inspected manually to remove body-methylated genes with significant non-CG methylation and to annotate additional genic regions that contain methylation only in the CG context (*Figure 6—figure supplement 2B*).

To identify regions of the genome that contain hyper-methylated DMRs, irrespective of their sequence context (*Figure 6D*), a set of merged hyper mC DMRs were generated as follows. First all hyper DMRs of the same context that were within 300 bp were merged into a single region using the mergeBed function with the –d 300 option. Then these merged CG, CHG, and CHH DMRs were combined using the bedops –everything function and finally, directly adjacent regions were joined

using the mergeBed function. Merged DMRs for the relaxed set of DMRs common between the *mbd7* and *lil* datasets as well as for the *rdd* datasets are available as part of *Supplementary file 2V and 2W* and were used for *Figure 6D* and *Figure 6—figure supplement 2D*.

## Heatmaps

Heatmaps showing the levels of methylation across individual DMRs were generated using the HOMER (Hypergeometric Optimization of Motif EnRichment) suite of genomics tools (*Heinz et al., 2010*). Homer 'TagDirectory' files cataloging the DNA methylation levels in the CG, CHG, and CHH contexts for each MethylC-seq experiment were generated using the wiggle files generated during the MethylC-seq mapping process (see **Genome-wide MethylC-seq mapping and quantification of DNA methylation levels**) at a precision level of three decimals (-precision 3). The methylation level over each DMR was then calculated using the annotatePeaks.pl script with the following options (none -ratio -noadj -size given -nogene -len 1 –ghist). Heatmaps were generated using Cluster (*de Hoon et al., 2004*) with the following options (-m a -g 4 -e 0) and were visualized in Java Tree-view. The data is either represented in a clustered form (*Figure 6—figure supplement 2A*) or an unclustered form (*Figure 6C* and *Figure 6—figure supplement 2C*). For the unclustered data, the DMRs were ordered based on the difference in the average values for the mutants and their controls (e.g. in *Figure 6C* the averaged difference was calculated using this equation [(*YJ_lil-1*_r1 + *YJ_lil-2*_r2 + *YJ_lil-5*_r5 + *YJ_mbd7-3*_r1 + *YJ_mbd7-4*_r1)/5] – [(*YJ*_r1 + *YJ*_r2 + *YJ*_r5)/3] and the rows were sorted largest to smallest).

## Library construction for mRNA-seq, data processing and identification of differentially expressed genes

For Col, *mbd7-3* and *mbd7-4*, RNA-seq libraries were generated using 2 µg of DNaseI-treated RNA and the NEBNext Ultra RNA Library Prep Kit for Illumina (New England Biolabs, E7530) according to the manufacturer's instructions. The libraries were pooled and then sequenced and processed by the Next Generation Sequencing Core at the Salk Institute for Biological Studies.

For *YJ, YJ lil-1*, *LUCH*, and *LUCH lil-2*, 10-day-old seedlings were used for RNA extraction using Trizol (Invitrogen, 15596–018), and the extracted RNA was treated with DNase I (Roche, 04716728001). 2 ug of the DNase I-treated RNA were used for RNA-seq library construction with the TruSeq RNA Sample Preparation Kit v2 (Illumina, FC-122–1002). The libraries were sequenced on an Illumina HiSeq 2000 instrument at the genomics core facility at UC Riverside. Image analysis and base calling were performed using the standard Illumina pipeline, version RTA 1.13.48.

For all samples, only reads that passed the Illumina quality control steps were included in subsequent analyses, and reads with multiple copies were considered as a single read for the mapping procedure. The reads were mapped to the TAIR10 *Arabidopsis* genome using TopHat v2.0.4 with default settings (*Kim et al., 2013*). Reads that mapped to multiple regions were discarded. The number of reads mapped to each gene was counted using a Perl script. Differentially expressed genes were identified using the R package edgeR (*Robinson et al., 2010*) from BioConductor (http://www.bioconductor.org). The false discovery rate (FDR) <= 0.05 and fold change >= 2 were used as the cutoff. For *Figure 6—figure supplement 4* the FPKM values were determined using the Homer analyzeRepeats.pl script using the -fpkm option for normalization.

## MBD7 ChIP

For the MBD7 ChIP-qPCR assays, tissue from F1 hybrids between the *YJ* and MBD7-GFP (*Wang et al., 2015*) lines was used and the ChIP assays were preformed following previously described procedures (*Liu et al., 2011*). Briefly, 5 g of 10-day-old seedlings were first ground in liquid nitrogen and then crosslinked in 1% formaldehyde (Amresco) for 10 min on ice. The chromatin was fragmented to 500 ~ 800 bp by sonication and the lysate was pre-cleared with 100 µl protein A agarose beads (Roche) for two hours before incubation with either no antibody or anti-GFP (Abcam, ab290) overnight at 4°C. Crosslinking was reversed by incubation at 65°C for 8 hr, afterwards, the DNA was purified with columns from the Qiagen plasmid extraction kit (Qiagen, 27106). Real-time PCR was conducted using input, no antibody control and antibody bound DNA in triplicates. Three biological repeats were performed to ensure reproducibility. All primers used in the ChIP-qPCR are listed in *Supplementary file 4*.

## Accession numbers

Genomic sequences reported in this manuscript have been submitted to NCBI GEO (http://www. ncbi.nlm.nih.gov/geo): gene expression, small RNA and DNA methylation data are under accession numbers GSE83557, GSE59639, and GSE83355, respectively.

## Acknowledgements

We thank Steven E Jacobsen (University of California, Los Angeles and HHMI) as well as Robert J Schmitz (University of Georgia) and Joseph R Ecker (Salk Institute and HHMI) for early support of the project and helpful comments and discussions. We also thank Rosa Castanon (Ecker lab, Salk Institute) for assistance with the MethylC-seq libraries, Francisco J Uribe and Jose Pruneda-Paz (University of California, San Diego) for assistance with luciferase imaging, Maggie Goodson (Law lab, Salk Institute) for technical assistance and members of the Ecker and Chory labs (Salk Institute and HHMI) for helpful discussion and the use of shared resources. Finally, we apologize for the omission of multiple gene names apart from the discussion section.

## Additional information

### Funding

| Funder | Grant reference number | Author |
|---|---|---|
| China Scholarship Council | | Dongming Li |
| Glenn Center for Aging Research at the Salk Institute | | Ana Marie S Palanca |
| Leona M. and Harry B. Helmsley Charitable Trust | | Ana Marie S Palanca Julie A Law |
| National Institutes of Health | P30 014195 | Ana Marie S Palanca Julie A Law |
| National Academy of Agricultural Science | PJ008725 | So Youn Won |
| National Institutes of Health | GM089778 | James A Wohlschlegel |
| National Natural Science Foundation of China | 30970265 | Beixin Mo |
| National Natural Science Foundation of China | 31210103901 | Beixin Mo |
| Gordon and Betty Moore Foundation | GBMF3046 | Xuemei Chen |
| National Institutes of Health | GM061146 | Xuemei Chen |
| National Natural Science Foundation of China | 91440105 | Xuemei Chen |
| Guangdong Innovation Research Team Fund | 2014ZT05S078 | Xuemei Chen |
| National Institutes of Health | GM112966 | Julie A Law |

The funders had no role in study design, data collection and interpretation, or the decision to submit the work for publication.

### Author contributions

DL, AMSP, SYW, Conceptualization, Formal analysis, Investigation, Writing—original draft, Writing—review and editing; LG, YF, AAV, LL, YZ, XL, XW, SLi, GY, Formal analysis, Investigation; BL, Formal analysis, Visualization; YJK, Resources, Investigation; SLi, Formal analysis, Supervision, Investigation; JL, JAW, HG, BM, Conceptualization, Supervision, Funding acquisition; XC, Conceptualization, Supervision, Funding acquisition, Writing—original draft, Writing—review and editing; JAL, Conceptualization, Data curation, Supervision, Funding acquisition, Investigation, Writing—original draft, Writing—review and editing

**Author ORCIDs**

Ana Marie S Palanca, http://orcid.org/0000-0003-4527-7604
Xuemei Chen, http://orcid.org/0000-0002-5209-1157
Julie A Law, http://orcid.org/0000-0001-7472-7753

## Additional files

### Supplementary files

• Source code 1. Custom Perl script used to trim 3' adapter sequences 839 from raw reads.

• Supplementary file 1. MethylC-seq and smRNAseq data processing (A) *LUC* reporter MethylC-seq mapping information. (B) *LUC* reporter MethylC-seq coverage. (C) *LUC* reporter mapping and coverage of small RNA data. (D) List of supplemental materials for *LUC* Reporter Genomics. (E) TAIR10 genome mapping and coverage information. (F) List of supplemental materials for the genome-wide analyses (TAIR10).

• Supplementary file 2. DMRs and DMR overlaps (A) Hyper and Hypo DMRs in the CG, CHG and CHH contexts. (B-F) Direct DMR overlaps in the various *lil* and *mbd7* datasets corresponding to 2-way, 3-way, 4-way, 5-way, and 6-way overlaps, respectively. (G-J) six way DMR coordinates in the hyper CG, CHG, CHH and hypo CG contexts, respectively. (K-P) Master DMR coordinates in the hyper CG, CHG, CHH, and hypo CG, CHG, and CHH contexts, respectively. (Q-U) Relaxed DMR coordinates in the hyper CG, CHG, CHH, and hypo CG, CHG, and CHH contexts, respectively. (V) *mbd7* and *lil* hyper DMRs, all mC contexts merged. (W) *rdd* hyper DMRs, all mC contexts merged.

• Supplementary file 3. Analysis of *lil* and *mbd7* RNAseq experiments.

• Supplementary file 4. Primers.

### Major datasets

The following datasets were generated:

| Author(s) | Year | Dataset title | Dataset URL | Database, license, and accessibility information |
|---|---|---|---|---|
| Dongming Li, Ana Marie S Palanca, So Youn Won, Lei Gao, Ying Feng, Ajay A Vashisht, Li Liu, Yuanyuan Zhao, Xigang Liu, Xiuyun Wu, Shaofang Li, Brandon Le, Yun Ju Kim, Guodong Yang, Shengben Li, Jinyuan Liu, James A Wohlschlegel, Beixin Mo, Xuemei Chen, Julie A Law | 2017 | The MBD7 complex promotes expression of methylated transgenes without significantly altering their methylation status | https://www.ncbi.nlm.nih.gov/geo/query/acc.cgi?acc=GSE83355 | Publicly available at the NCBI Gene Expression Omnibus (accession no: GSE83355) |
| Dongming Li, Ana Marie S Palanca, So Youn Won, Lei Gao, Ying Feng, Ajay A Vashisht, Li Liu, Yuanyuan Zhao, Xigang Liu, Xiuyun Wu, Shaofang Li, Brandon Le, Yun Ju Kim, Guodong Yang, Shengben Li, Jinyuan Liu, James | 2017 | The MBD7 complex promotes expression of methylated transgenes without significantly altering their methylation status | https://www.ncbi.nlm.nih.gov/geo/query/acc.cgi?acc=GSE59639 | Publicly available at the NCBI Gene Expression Omnibus (accession no: GSE59639) |

A Wohlschlegel,
Hongwei Guo,
Beixin Mo, Xuemei
Chen, Julie A Law

| | | | | |
|---|---|---|---|---|
| Dongming Li, Ana Marie S Palanca, So Youn Won, Lei Gao, Ying Feng, Ajay A Vashisht, Li Liu, Yuanyuan Zhao, Xi-gang Liu, Xiuyun Wu, Shaofang Li, Brandon Le, Yun Ju Kim, Guodong Yang, Shengben Li, Jinyuan Liu, James A Wohlschlegel, Hongwei Guo, Beixin Mo, Xuemei Chen, Julie A Law | 2017 | The MBD7 complex promotes expression of methylated transgenes without significantly altering their methylation status | http://www.ncbi.nlm.nih.gov/geo/query/acc.cgi?acc=GSE83557 | Publicly available at the NCBI Gene Expression Omnibus (accession no: GSE83557) |

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
