## [Decision Letter]

Thank you for submitting your article "The MBD7 complex promotes expression at methylated transgenes without significantly altering their methylation status" for consideration by *eLife*. Your article has been reviewed by three peer reviewers, and the evaluation has been overseen by Jessica Tyler as the Senior Editor and Steven Henikoff as Guest Reviewing Editor. The following individual involved in review of your submission has agreed to reveal their identity: Robert A Martienssen (Reviewer #2).

The reviewers have discussed the reviews with one another and the Reviewing Editor has drafted this decision to help you prepare a revised submission.

Summary:

This report describes a forward genetic screen for Arabidopsis anti-silencers that identifies the protein LIL, which through yeast 2-hybrid analysis leads to the finding that MBD7 and associated factors are required for the expression of two luciferase reporter transgenes. Several papers have recently been published on MBD7, which concluded that it functions to promote DNA demethylation. The data presented here lead to a very different conclusion, that MBD7 does not promote DNA demethylation. Although this paper does not determine how the MBD7 complex is affecting transgene expression, it was felt that this study brings compelling evidence for a new mechanism of the MBD7-LIL complex in transgene regulation.

The Reviewing Editor and other reviewers are generally interested in your work and find the study of significant importance to be suitable for publication in *eLife*. However, the consensus among the reviewers is that the following aspects of the paper must be strengthened before we proceed further.

Essential revisions:

Below are the points that should be addressed upon revision. Point 1 is a required experiment, and Point 2 is a suggested experiment. Points 3-11 may require textual changes and/or additional analyses, but should not require additional experimental evidence. Whether or not you agree with a point, please address it in a point-by-point response.

1) Does MBD7 bind to the methylated promoter of the reporter gene? A ChIP-qPCR experiment would confirm that.

2) It is suggested that you choose candidate genes that came up positive in ChIP in previous studies and assay for a mutant effect on expression in seed, where the MBD7 complex is highly expressed, and show that the effect is independent of methylation. However, given the complexity of seed tissue, we recognize that this kind of experiment might be challenging to interpret.

3) Clarifications are needed for understanding what was done in the experiments described in Figure 1—figure supplement 1 and Figure 4.

4) With respect to the insufficiency of d35S promoter 3'-end hypermethylation to cause LUC gene silencing, the possibility needs to be considered that CHH methylation alone is sufficient to cause silencing.

5) With respect to the inability of a LIL transgene not complementing the identified methylation differences, the arguments presented should take into account the partial complementation seen in Figure 6—figure supplement 3).

6) Can downstream function of the MBD7 complex be distinguished from independent function? These distinct possibilities need to be explicitly discussed.

7) Interaction affinities, if they have been determined experimentally, should be compared between the previously published Y2H analysis and the current one.

8) Subtle changes in the transgene methylation might have gone undetected.

9) Global misregulation of methylation caused by loss of the MBD7 complex might be detected by looking an increase in variation of methylation in the mutants.

10) Could there be redundancy with other MBD-ACD proteins?

11) Given the published results of MBD7 binding methylated genomic regions and influencing transcription at a similar time point as the one tested here, you should comment on these discrepancies.

*Reviewer #1:*

This manuscript identifies MBD7 and associated factors as required for the expression of two LUC transgenes. Several papers have recently been published on the function of MBD7 and interacting proteins (Lang et al., 2015; Li et al., 2015; Want et al., 2015). These papers concluded that MBD7 functions at ROS1 targets to promote DNA demethylation. This is despite the fact that the overlap between *ros1 dml2 dml3* hypermethylated regions and *mbd7* hypermethylated regions is actually quite low. The data presented here are similar, but have the benefit of extensive replication. A very different conclusion is reached – that MBD7 does not promote DNA demethylation. Although this paper does not determine how the MBD7 complex is affecting transgene expression, this is an important contribution to understanding the function (or excluding potential functions) of α-crystallin domain and MBD7 proteins.

Specific comments to address:

For the LUCH and YJ methylation profiles in Figure 1—figure supplement 1, how were reads mapped to the nearly identical d35S sequences upstream of LUC and NPTII? How were 24nt sRNAs uniquely mapped? It is essential to know whether or not the LUC gene has promoter methylation, as the system is described as one where genes have high promoter methylation but are expressed. (This may be in the legend but it was cut-off.) This also applies to Figure 4. It seems that this is eventually addressed in Figure 5 for YJ, but providing additional clarity in the main text is recommended. In Figure 1—figure supplement 1 please also show the methylation profiles using uniquely-mapping reads only. It remains unclear whether LUCH has a hypermethylated promoter.

In the subsection “Genetic uncoupling of the hyper-methylation and LUC expression phenotypes at the YJ reporter” the authors conclude that hypermethylation of the 3' end of the d35S promoter is not sufficient to cause LUC gene silencing. This is because *nrpe1* and *ago4* mutants are also hypermethylated, yet LUC expression is higher. *nrpe1* and *ago4* mutants only have higher methylation in the CG context, and have lower CHH methylation than the other genotypes. The counter-argument is that CHH methylation alone is sufficient to cause LUC gene silencing. The authors should modify this conclusion to be more inclusive of the data presented.

The arguments about a LIL transgene not complementing the identified methylation differences (subsection “Genome-wide profiling shows that MBD7 and LIL have minimal effects on DNA methylation”, third paragraph) are not fully convincing because the LIL transgene does not appear to fully complement the LUC expression phenotype (Figure 6—figure supplement 3).

Why do the authors conclude that the MBD7 complex functions downstream of DNA methylation (subsection “Genome-wide profiling shows that MBD7 and LIL have minimal effects on DNA methylation”, last paragraph and others)? Functioning entirely independent of methylation seems as likely. Independence is occasionally mentioned, but seem to be used interchangeably with downstream in the text, when biologically these would seem to mean two very different things. Just because the YJ and LUCH transgenes are methylated does not mean that the complex exclusively regulates expression of methylated regions of the genome. There may be some other feature about these transgenes that causes the complex to act on them.

*Reviewer #2:*

The study presented here reports a forward genetic screen for anti-silencing factors identifying LIL, an Arabidopsis gene also known as IDM3/IDL1. Through Y2H and Co-IP experiments, the authors recover the different interacting partners of LIL, highlighting a putative complex containing MBD7, ROS4/IDM1 and ROS5/IDM2. This particular complex is thought to participate in the recruitment of the ROS1 protein to DNA so it can excise methylated cytosine thereby promoting transcription (Lang et al., 2015; Li et al., 2015; Wang et al., 2015). The present work suggests that the MBD7-LIL complex promotes transgene transcription independently of any variation of DNA methylation, contrasting with previous studies. For this, the authors thoroughly examine the hyper-methylation phenotype of the mutants and conclude that it is inconsequential for the transgene silencing and barely significant genome-wide when comparing between replicates and controls. While this study brings compelling evidence for a new mechanism of the MBD7-LIL complex in transgene regulation, I would suggest the two following points be addressed before considering publications:

1) Since the most important conclusion for this paper comes from two independent transgenes, it would be important to demonstrate the binding of MBD7 to the methylated promoters of at least one of these reporter genes. Since the authors already have introduced tagged versions of MBD7 in their YJ reporter line, a ChIP-qPCR could easily confirm a direct role for the complex in promoting transgene expression.

2) A genome-wide ChIP experiment has revealed thousands of binding sites for MBD7 in transposable elements and genes (Lang et al., 2015). Yet in the present study, RNAseq experiment fails to identify any target in *mbd7* and *lil* mutants. To prove the biological relevance of their proposed methylation-independent promotion of transcription, it would be interesting to investigate changes in expression further. Since the genes involved in the MBD7 complex appear to have high expression in seed (according to eFP browser and genevestigator), targeted expression survey (based on loci identified in previous studies) could be conducted rapidly in these tissues for at least one mutant background. It would also be essential to show that the changes in expression are independent of DNA methylation changes.

*Reviewer #3:*

Recent publications indicate that the MBD7 and IDM3 enhance transcription through demethylation. This manuscript present contradictory data that suggest these components instead activate transcription from methylated targets without altering methylation. While this alternative viewpoint is important to consider and the MethylC-seq analysis is very compelling, I am not convinced that the authors have ruled out alternative explanations. In particular, issues of redundancy between MBD-ACD complexes is not addressed, nor is there evidence for transcriptional activation of endogenous loci in the absence of MBD7-IDM3. As is often the case when attempting to argue against published conclusions, the data are extensive and complex, and might not be appropriate for a general scientific publication.

Major comments:

Because the MBD7d3 construct does contain a tiny overlap with the StkC domain, it remains possible that this bit of StkC is mediating interaction with IDM3. Since the MBD7-IDM3 interaction was previously mapped to the StkC domain with Y2H, the authors should include that construct in their analysis to directly compare interaction affinities.

Although MethylC-seq and Bisulfite/sanger sequencing indicate that there is little to no change in methylation of the YJ transgene, it remains possible that the subtle change in methylation of one small region is responsible for the observed change in transcription. Unfortunately, the *nrpe1* and *ago4* mutants do not completely rule out this possibility due to the possible "compensatory effects" that the authors acknowledge (subsection “Genetic uncoupling of the hyper-methylation and LUC expression phenotypes at the YJ reporter”, last paragraph).

Is there the possibility that loss of the MBD7 complex causes misregulation of methylation such to increase variation in methylation across the genome? This could account for the large number of DMRs identified in each dataset that were not conserved between replicates. With so many control samples, it should be possible to determine the variation in the controls and compare it with the variation in the mutants to detect any possible increase in variation.

If the MBD7 complex functions as the authors suggest – countering DNA methylation to allow transcription, then why aren't genes differentially expressed in the *lil* and *mbd7* backgrounds (subsection “Genome-wide profiling shows that MBD7 and LIL have minimal effects on DNA methylation”, third paragraph)? I'm afraid that this might be a fatal flaw – the assertion that MBD7 regulates transcription without changing methylation is based on a lack of changes in DNA methylation, but there is a similar lack of change in gene expression.

The authors offer no evidence that MBD-ACD don't function redundantly at most genomic loci, making it difficult to agree with the authors' key conclusion at the end of the first paragraph of the subsection “The MBD7 complex and the DNA demethylation machinery”.

---

## [Author Response]

*Essential revisions:*

*Below are the points that should be addressed upon revision. Point 1 is a required experiment, and Point 2 is a suggested experiment. Points 3-11 may require textual changes and/or additional analyses, but should not require additional experimental evidence. Whether or not you agree with a point, please address it in a point-by-point response.*

*1) Does MBD7 bind to the methylated promoter of the reporter gene? A ChIP-qPCR experiment would confirm that.*

To address this question we carried out numerous ChIP-qPCR and ChIP-seq experiments using different transgenic lines expressing tagged versions of MBD7 or LIL under the control of their endogenous promoters (e.g. pMBD7::MBD7-3xHA, pMBD7::MBD7-3xFlag, and pLIL::9xMyc-LIL). Despite extensive optimization of different experimental conditions (in vivo vs. in vitro crosslinking, Formaldehyde (FD) vs. ethylene glycol bis succinimidyl sussinate (EGS) +FD crosslinking, tissue quantities up to ~5 g, and several published ChIP protocols), no enrichment was observed either at the *YJ* reporter or at other genomic loci. However, using a previously published, GFP tagged MBD7 construct driven by a 35S promoter, we were able to observe modest, but consistent and reproducible enrichment of MBD7 at the *YJ* reporter, specifically at the methylated *d35S* promoter driving *LUC* expression. We have now added this data as Figure 4 and have added the following text to the Results section:

“As these findings reveal a correlation between the presence of a functional MBD7 complex and the DNA methylation status of the YJ reporter, several chromatin immunoprecipitation (ChIP) experiments were conducted to determine whether the MBD7 complex associates with this reporter. […] Taken together, these methylation and ChIP analyses are consistent with the hyper-methylation phenotype observed in the methyl-cutting assays and suggest a direct role for the MBD7 complex in regulating expression at the *YJ* reporter.”

*2) It is suggested that you choose candidate genes that came up positive in ChIP in previous studies and assay for a mutant effect on expression in seed, where the MBD7 complex is highly expressed, and show that the effect is independent of methylation. However, given the complexity of seed tissue, we recognize that this kind of experiment might be challenging to interpret.*

We thank the reviewer for this very nice suggestion, however as detailed below, we were not able to identify any target genes with altered gene expression in seeds.

As suggested, we collected a list of 23 candidate genes that were either shown to be mis-expressed or to be bound by MBD7 in previous publications. After testing each primer set on gDNA, RNA was extracted from dry seeds to make cDNA. Of the 23 candidate genes only 8 were expressed in dry seeds and none showed altered expression in *lil* and *mbd7* mutants.

Author response image 1.Expression of putative MBD7 targets in dry seeds.Quantification of transcript levels by RT-qPCR using pooled dry seeds. Transcript levels were normalized to *UBIQUITIN5* with the expression level of the target loci in the *YJ* and *LUCH* controls set to one. Error bars indicate the standard deviation from at least two biological replicates.**DOI:**
http://dx.doi.org/10.7554/eLife.19893.026

*3) Clarifications are needed for understanding what was done in the experiments described in Figure 1—figure supplement 1 and Figure 4.*

We thank the reviewers for calling this issue to our attention. We agree that it is important to know how multi-mapping reads are treated, and indeed had included statements addressing this point at the ends of Figure 1—figure supplement 1 and Figure 4 legends, but the legend for Figure 1—figure supplement 1 was cut off calling this issue to our attention. The following statement is now present in both legends “Note that two *d35S* promoters driving *LUC* and *NPTII* are 94% identical in sequence, thus the DNA methylation data includes both multi-mapping and unique reads”.

We have also made this point more clear in the Methods, under the “MethylC-seq analysis at the *YJ* and *LUCH* transgenes” section, by including the following sentence: “Since the *d35S* promoters driving the expression of the *LUC* and *NPTII* genes are 94% identical, both unique and multi-mapping reads were included, with the maximum number of equal best hits set to 2 (-w 2).”

In addition, as suggested by reviewer 1, we have also included in Figure 1—figure supplement 1 the DNA methylation and 24nt siRNA profiles allowing only uniquely mapped reads. Along with this new data, we have added the necessary information to the Methods section and to [Supplementary-material SD3-data] and [Supplementary-material SD4-data].

Finally, as also suggested by reviewer 1, we further altered the main text in two places to further clarify issues posed by the 94% identical *d35S* promoters. First, in the section describing the characterization of the two reporters, we made the following alteration:

“Given the known role of DNA methylation in regulating *LUC* expression at the *LUCH* reporter [Won et al., 2012], the DNA methylation and siRNA profiles for both *LUCH* and *YJ* reporters were determined by MethylC-sequencing (MethylC-seq) and small RNA sequencing (smRNA-seq), respectively ([Supplementary-material SD3-data]), allowing either multi-mapping (Figure 1—figure supplement 1) or unique reads (Figure 1—figure supplement 1).”

Second, in the section detailing the traditional bisulfite sequencing, we made the following alteration to emphasize that such analysis is required to assess the methylation specifically at the *d35S* promoter driving *LUC* expression, independently of both the 94% identical *d35S* promoter driving *NPTII* expression in the YJ transgene and similar *d35S* promoters in the T-DNA mutant lines.

“Thus, to specifically assess DNA methylation at the *d35S* promoter driving the *LUC* gene, without interference from the 94% identical *d35S* promoter driving *NPTII* expression (Figure 1—figure supplement 1) or the similar *d35S* promoters present in the T-DNA insertion mutant backgrounds, traditional bisulfite conversion assays coupled with Sanger sequencing were conducted.”

“These findings demonstrate that the two methods of bisulfite sequencing are comparable and, unlike the MethylC-seq data (Figure 1—figure supplement 1), definitively show that the changes in DNA methylation observed in the *lil* and *mbd7* mutants occur at the *35S* promoter driving *LUC* expression.”

*4) With respect to the insufficiency of d35S promoter 3'-end hypermethylation to cause LUC gene silencing, the possibility needs to be considered that CHH methylation alone is sufficient to cause silencing.*

We agree that this is an important point. Thus, we have modified the main text to soften our conclusion, as shown below, and have included a more detailed visual representation of the traditional bisulfite sequencing using cymate, which we hope will aid in the data interpretation. However, we would like to raise the point that if CHH methylation alone was enough to account for all the gene silencing, then one would expect higher LUC expression in the *YJ nrpe1* and *YJ ago4* backgrounds as compared to the YJ control, which is not what we observed. Thus, a more complicated compensatory model would need to be invoked.

“While compensatory effects on gene expression due to decreased non-CG methylation at other regions of the *d35S* promoter in the *ago4* and *nrpe1* mutants cannot be fully excluded, these findings suggest that the hyper-methylation at the 3’ end of the *d35S* promoter region alone is not sufficient to cause gene silencing.”

“Secondly, we demonstrated that hyper-methylation at the 3’ end of the *d35S* promoter in the *YJ* reporter does not appear to be sufficient to cause gene silencing (Figure 5). As such, the MBD7 complex joins a small number of factors including MOM1, MORC1, MORC6, ATXR5, and ATXR6, that can act primarily downstream of DNA methylation.”

“Furthermore, we found that while mutations in components of the MBD7 complex led to increased DNA methylation at the d35S promoter driving LUC expression, this methylation alone did not appear to be sufficient to cause LUC silencing.”

“Conversely, all three mutants (*ago4-6, nrpe1-11* and *rdd*) showed a hyper-methylation phenotype similar to that observed in the *lil* and *mbd7* mutants at the 3’ end of the *35S* promoter (Figure 5 and Figure 5—figure supplement 1).”

*5) With respect to the inability of a LIL transgene not complementing the identified methylation differences, the arguments presented should take into account the partial complementation seen in Figure 6—figure supplement 3).*

We agree that the level of complementation should be taken into account when interpreting the MethylC-seq data. Therefore, we have added the following sentence to the main text.

“In general, the complementing lines more closely resemble the *YJ lil-1_r7* mutant line than the *YJ_r7* control (Figure 6), suggesting minimal complementation overall. Although the possibility of partial complementation at some loci cannot be excluded, even at the most robust set of DMRs, only 1 of 33 DMRs returned to control levels of DNA methylation (Figure 6—figure supplement 1, hyper-CHH site 1).”

*6) Can downstream function of the MBD7 complex be distinguished from independent function? These distinct possibilities need to be explicitly discussed.*

We thank the reviewers for pointing out that we were not clear enough in our usages of the terms “downstream” and “independent”. Based on the 5 AZA experiments presented in Figure 3 we have shown that, at least at the *YJ* and *LUCH* transgenes, the anti-silencing functions of the MBD7 complex are dependent on DNA methylation (i.e. expression is only lower in the *mbd7* and *lil* mutants when methylation is present). Thus, we conclude that they act downstream rather than independently of DNA methylation at these loci. However, in several other places in the main text we state that the MBD7 complex function/acts in a manner that is “largely independent of changes in DNA methylation”. In hindsight, we now appreciate that this caused confusion, and have reworded the following statements for clarity.

“Given the absence of endogenous targets transcriptionally regulated by the MBD7 complex, it is likely that this complex functions redundantly with other MBD-ACD complexes and/or is required only under specific conditions. Nonetheless, our findings demonstrate that this complex has the ability to regulate gene expression in a manner largely downstream of DNA methylation at *LUC* reporters.”

“Rather than functioning as part of the DNA demethylation pathway, as has been previously hypothesized [Lang et al., 2015; Li et al., 2015; Wang et al., 2015], our results suggest that the MBD7 complex functions in a manner largely downstream of DNA methylation. As very few proteins have been characterized that function downstream of DNA methylation, further characterization of this complex…”.

“Further support for the notion that the primary function of the MBD7 complex is downstream of DNA methylation comes from a comparison of the reporter transgenes used to identify components of this complex.”

*7) Interaction affinities, if they have been determined experimentally, should be compared between the previously published Y2H analysis and the current one.*

We have not determined interaction affinities.

*8) Subtle changes in the transgene methylation might have gone undetected.*

We agree that we cannot exclude the possibility that subtle changes in methylation could play important roles in regulating gene expression. To further address this question we have now taken the suggestion of reviewer 2 and added a visual representation of all the traditional bisulfite sequencing as a new figure (Figure 5—figure supplement 1). Providing this data should make it easier to interpret the data presented here and aid in future re-analysis of the data as the field progresses and we gain a deeper understanding of how DNA methylation controls gene expression.

*9) Global misregulation of methylation caused by loss of the MBD7 complex might be detected by looking an increase in variation of methylation in the mutants.*

We thank the reviewer for raising this interesting point. However, as there are many ways to both define and assess variation in methylation, this is a quite difficult question to definitively address. To begin addressing this question, we did investigate the variation in the control vs experimental samples by looking at the variation in the number of DMRs amongst the sample sets. The data is plotted below:

Author response image 2.Investigation into variation in DNA methylation.Comparisons of the average number of DMRs observed between all pairwise combinations of three YJ replicates (YJ_r1, YJ_r5 and YJ_r7) or between all pairwise combinations of these controls and their corresponding YJ_lil data sets. Error bars represent the standard deviation as a measure of variance.**DOI:**
http://dx.doi.org/10.7554/eLife.19893.027

While the variance (std dev error bars) seems to be similar between the wildtype and mutant comparisons in all cases, suggesting there is no increase in the methylation variation in the mutants, we cannot confidently say whether there are statistically significant differences because our sample size is too small to determine whether or not the DMR numbers follow a particular distribution from which p values can be determined. Thus, we feel it would be premature to make statements regarding methylation variation based on our current datasets. This is especially true as no proteins or pathways have been identified that serve to increase the variation in methylation patterns and thus the burden of proof for such a claim would be very high.

*10) Could there be redundancy with other MBD-ACD proteins?*

We agree with the assessment that MBD-ACD proteins/complexes could function redundantly when it comes to regulating endogenous loci and thank the reviewers for calling to our attention that we should emphasize this more clearly in the manuscript. In its original form, this possibility was mentioned twice in the Discussion, but not mentioned in the Results section. We have now edited and/or added the following statements in the Results to more clearly acknowledge the possibility of redundant functions between the various MBD and ACD proteins at endogenous loci.

“Thus, although there are clear genetic connections between MBD7, LIL and ROS1 in the regulation of several transgenic reporters [Lang et al., 2015; Li et al., 2015; Wang et al., 2015], as a general rule, the demethylation pathway does not appear to function in a manner that depends solely on either MBD7 or LIL.”

“Given the absence of endogenous targets transcriptionally regulated by the MBD7 complex, it is likely that this complex functions redundantly with other MBD-ACD complexes and/or is required only under specific conditions. Nonetheless, our findings demonstrate that this complex has the ability to regulate gene expression in a manner largely downstream of DNA methylation at LUC reporters.”

“Thus, while we cannot fully rule out a role for the MBD7 complex as a highly locus-specific regulator of the DNA demethylation pathway, nor do we know the extent to which other MBD-ACD complexes might function redundantly with the MBD7 complex and mask its role at endogenous targets and/or connections with the demethylase machinery, we currently favor a model based on our extensive transgene analysis in which the MBD7 complex functions largely downstream of DNA methylation to promote gene expression through a yet unknown mechanism.”

*11) Given the published results of MBD7 binding methylated genomic regions and influencing transcription at a similar time point as the one tested here, you should comment on these discrepancies.*

To address this comment we have now added a new supplemental figure (Figure 6—figure supplement 4) and a separate section to the Results in which we provide additional details regarding our RNAseq data and also draw comparisons with the genes identified in previous studies. See below:

“Transcriptome profiling of *mbd7* and *lil* mutants

To further investigate the role of the MBD7 complex in gene regulation, we assessed the effects of the *lil* and *mbd7* mutants on gene expression by transcriptome profiling. […] However, perhaps differences in growth conditions and/or the differing sensitivities and normalization procedures of the assays used to assess gene expression levels (i.e. mRNA-seq vs qPCR), as well as the already low expression levels of the downregulated genes under our conditions, represent contributing factors (Figure 6—figure supplement 4).”

*Reviewer #1:*

*[…] Specific comments to address:*

*For the LUCH and YJ methylation profiles in Figure 1—figure supplement 1, how were reads mapped to the nearly identical d35S sequences upstream of LUC and NPTII? How were 24nt sRNAs uniquely mapped? It is essential to know whether or not the LUC gene has promoter methylation, as the system is described as one where genes have high promoter methylation but are expressed. (This may be in the legend but it was cut-off.) This also applies to Figure 4. It seems that this is eventually addressed in Figure 5 for YJ, but providing additional clarity in the main text is recommended. In Figure 1—figure supplement 1 please also show the methylation profiles using uniquely-mapping reads only. It remains unclear whether LUCH has a hypermethylated promoter.*

As detailed in Essential revisions point 3, we have now included data showing methylation and siRNA levels at the YJ and LUCH reporters using only uniquely mapping reads and have further clarified the main text.

*In the subsection “Genetic uncoupling of the hyper-methylation and LUC expression phenotypes at the YJ reporter” the authors conclude that hypermethylation of the 3' end of the d35S promoter is not sufficient to cause LUC gene silencing. This is because nrpe1 and ago4 mutants are also hypermethylated, yet LUC expression is higher. nrpe1 and ago4 mutants only have higher methylation in the CG context, and have lower CHH methylation than the other genotypes. The counter-argument is that CHH methylation alone is sufficient to cause LUC gene silencing. The authors should modify this conclusion to be more inclusive of the data presented.*

As detailed in Essential revisions point 4, we have amended the text to address this point.

*The arguments about a LIL transgene not complementing the identified methylation differences (subsection “Genome-wide profiling shows that MBD7 and LIL have minimal effects on DNA methylation”, third paragraph) are not fully convincing because the LIL transgene does not appear to fully complement the LUC expression phenotype (Figure 6—figure supplement 3).*

As detailed in Essential revisions point 5, we have altered the main text to include the possibility that there could be some partial complementation.

*Why do the authors conclude that the MBD7 complex functions downstream of DNA methylation (subsection “Genome-wide profiling shows that MBD7 and LIL have minimal effects on DNA methylation”, last paragraph and others)? Functioning entirely independent of methylation seems as likely. Independence is occasionally mentioned, but seem to be used interchangeably with downstream in the text, when biologically these would seem to mean two very different things. Just because the YJ and LUCH transgenes are methylated does not mean that the complex exclusively regulates expression of methylated regions of the genome. There may be some other feature about these transgenes that causes the complex to act on them.*

As detailed in Essential revisions point 6, the 5 aza experiments in Figure 3 show that the function of the MDB7 complex is dependent on the presence of DNA methylation. This finding, along with the DNA methylation analyzes showing minimal changes in DNA methylation in the *mbd7* and *lil* mutants, lead us to conclude that the MBD7 complex functions largely downstream of DNA methylation. We have modified the main text as necessary to avoid confusion cause by use of the phrase “independent of changes in DNA methylation”.

*Reviewer #2:*

*[…] While this study brings compelling evidence for a new mechanism of the MBD7-LIL complex in transgene regulation, I would suggest the two following points be addressed before considering publications:*

*1) Since the most important conclusion for this paper comes from two independent transgenes, it would be important to demonstrate the binding of MBD7 to the methylated promoters of at least one of these reporter genes. Since the authors already have introduced tagged versions of MBD7 in their YJ reporter line, a ChIP-qPCR could easily confirm a direct role for the complex in promoting transgene expression.*

As detailed in Essential revisions point 1, we have added MBD7-GFP ChIP-qPCR data showing enrichment of this protein at the *YJ* reporter as Figure 4.

*2) A genome-wide ChIP experiment has revealed thousands of binding sites for MBD7 in transposable elements and genes (Lang et al., 2015). Yet in the present study, RNAseq experiment fails to identify any target in mbd7 and lil mutants. To prove the biological relevance of their proposed methylation-independent promotion of transcription, it would be interesting to investigate changes in expression further. Since the genes involved in the MBD7 complex appear to have high expression in seed (according to eFP browser and genevestigator), targeted expression survey (based on loci identified in previous studies) could be conducted rapidly in these tissues for at least one mutant background. It would also be essential to show that the changes in expression are independent of DNA methylation changes.*

As detailed in Essential revisions point 2, we were unable to show altered expression of candidate MBD7 targets in seeds from *lil* and *mbd7* mutants.

*Reviewer #3:*

*[…] Because the MBD7d3 construct does contain a tiny overlap with the StkC domain, it remains possible that this bit of StkC is mediating interaction with IDM3. Since the MBD7-IDM3 interaction was previously mapped to the StkC domain with Y2H, the authors should include that construct in their analysis to directly compare interaction affinities.*

While we agree that this is possible, it seems unlikely given the limited overlap (only 11 amino acids), which is why in the main text we stated that our findings “…suggesting that the interaction between MBD7 and LIL can be mediated by several regions of the MBD7 protein…”.

*Although MethylC-seq and Bisulfite/sanger sequencing indicate that there is little to no change in methylation of the YJ transgene, it remains possible that the subtle change in methylation of one small region is responsible for the observed change in transcription. Unfortunately, the nrpe1 and ago4 mutants do not completely rule out this possibility due to the possible "compensatory effects" that the authors acknowledge (subsection “Genetic uncoupling of the hyper-methylation and LUC expression phenotypes at the YJ reporter”, last paragraph).*

As detailed in Essential revisions point 8, we agree that we cannot exclude the possibility that subtle changes in methylation could play important roles in regulating gene expression and have included a more detailed representation of our DNA methylation data (Figure 5—figure supplement 1) to help address this issue.

*Is there the possibility that loss of the MBD7 complex causes misregulation of methylation such to increase variation in methylation across the genome? This could account for the large number of DMRs identified in each dataset that were not conserved between replicates. With so many control samples, it should be possible to determine the variation in the controls and compare it with the variation in the mutants to detect any possible increase in variation.*

As detailed in Essential revisions point 9, we feel it would be premature to make statements regarding methylation variation based on our current datasets and level of statistical expertise.

*If the MBD7 complex functions as the authors suggest – countering DNA methylation to allow transcription, then why aren't genes differentially expressed in the lil and mbd7 backgrounds (subsection “Genome-wide profiling shows that MBD7 and LIL have minimal effects on DNA methylation”, third paragraph)? I'm afraid that this might be a fatal flaw – the assertion that MBD7 regulates transcription without changing methylation is based on a lack of changes in DNA methylation, but there is a similar lack of change in gene expression.*

As detailed in Essential revisions point 10, we feel that redundancy with in MBD and LIL families could account for the lack of expression defects at endogenous loci and have modified the text as specified to make this point more clear.

*The authors offer no evidence that MBD-ACD don't function redundantly at most genomic loci, making it difficult to agree with the authors' key conclusion at the end of the first paragraph of the subsection “The MBD7 complex and the DNA demethylation machinery”.*

As detailed in Essential revisions point 10, we have now added a statement to the Results acknowledging that the MBD-ACD complexes may act redundantly at endogenous loci.